# Myofiber-specific TEAD1 overexpression drives satellite cell hyperplasia and counters pathological effects of dystrophin deficiency

Sheryl Southard[1†], Ju-Ryoung Kim[2,3,4,5†], SiewHui Low[1], Richard W Tsika[2,3,4,5*], Christoph Lepper[1*]

[1]Department of Embryology, Carnegie Institution for Science, Baltimore, United States; [2]Department of Biochemistry, University of Missouri, Columbia, United States; [3]School of Medicine, University of Missouri, Columbia, United States; [4]Department of Biomedical Sciences, University of Missouri, Columbia, United States; [5]College of Veterinary Medicine, University of Missouri, Columbia, United States

**Abstract** When unperturbed, somatic stem cells are poised to affect immediate tissue restoration upon trauma. Yet, little is known regarding the mechanistic basis controlling initial and homeostatic 'scaling' of stem cell pool sizes relative to their target tissues for effective regeneration. Here, we show that TEAD1-expressing skeletal muscle of transgenic mice features a dramatic hyperplasia of muscle stem cells (i.e. satellite cells, SCs) but surprisingly without affecting muscle tissue size. Super-numeral SCs attain a 'normal' quiescent state, accelerate regeneration, and maintain regenerative capacity over several injury-induced regeneration bouts. In dystrophic muscle, the TEAD1 transgene also ameliorated the pathology. We further demonstrate that hyperplastic SCs accumulate non-cell-autonomously via signal(s) from the TEAD1-expressing myofiber, suggesting that myofiber-specific TEAD1 overexpression activates a physiological signaling pathway(s) that determines initial and homeostatic SC pool size. We propose that TEAD1 and its downstream effectors are medically relevant targets for enhancing muscle regeneration and ameliorating muscle pathology.

*For correspondence: tsikar@ missouri.edu (RWT); lepper@ ciwemb.edu (CL)

†These authors contributed equally to this work

Competing interests: The authors declare that no competing interests exist.

## Introduction

Maintenance and repair of many tissues depend on reserve populations of tissue-specific stem cells. In the unperturbed state, these reserve cells are maintained at a defined pool size. Upon tissue damage, they expand to give rise to progenitors that can differentiate for efficient and timely tissue restoration as well as self-renew to re-establish the stem cell pool. With age, tissue function and regenerative capacity decline due to alterations in stem cell intrinsic function, microenvironment (niche), and systemic cues. Collectively, these changes lead to a decline in the number of functional stem cells (*Oh et al., 2014*). The skeletal muscle system provides an experimental paradigm for understanding stem cell-based tissue regeneration. Because of the essentiality of skeletal muscles for mobility, metabolic homeostasis, and thermogenesis, the clinical relevance of resident muscle stem cells cannot be over-stated.

Skeletal muscle tissue can mount effective tissue restoration even after up to 50 weekly myotoxin-induced injuries in the mouse (*Luz et al., 2002*). The muscle stem cells driving this remarkable life-long regenerative response are satellite cells (SCs), which were initially discovered by Alexander

**eLife digest** Skeletal muscles are primarily composed of cells called muscle fibers, which attach to bones via tendons. These muscle fibers contract to help move the body. Muscle also contains a population of muscle stem cells that repair injured tissue. Normally, in adult skeletal muscle, these stems cells are in a resting state. However, upon injury, the stem cells become activated, divide to increase in number and then develop into new muscle fibers to replace those that were damaged. The balance between the number of stem cells and the size of the muscle must be tightly regulated to ensure that there are enough stem cells to fully regenerate the tissue after injury. However, little is known about how tissues keep their number of stem cells in proportion with their overall size.

Previous attempts to make mice with more muscle stem cells invariably also created mice with larger muscles overall. This raised the question: is it possible to increase the numbers of stem cells without changing the size of the muscle?

Now, Southard, Kim et al. show it is possible and report that mice engineered to overproduce a protein called Tead1 in their muscle fibers have up to 6-times more stem cells yet normally sized muscles. Tead1 is a transcription factor that controls the activity of a number of genes as part of a major signaling pathway.

The stem cells in mice that overproduce Tead1 began to increase in number two weeks after the mice were born because they went through additional rounds of cell division before they entered the resting state. Further experiments then showed that having more stem cells meant that the muscles were repaired more quickly after an injury. Additionally, when mice with extra Tead1 had a mutation that normally leads to muscle wasting, experiments showed that the progression of the disease was stunted.

Southard, Kim et al. also show that the muscle fibers that are directly attached to the muscle stem cells are needed for the stem cells to increase in number in the Tead1-overexpressing mice. Together these findings suggest that a signal from the muscle fiber to its stem cells regulates the size of the stem cell population in the tissue. The next challenge is to uncover the molecule (or molecules) that signals from the muscle fiber to the stem cells and to gain deeper insight into how the Tead1 protein can counteract the effects of a muscle wasting disease.

Mauro in Xenopus (*Mauro, 1961*). SCs comprise a small population (5 to 7% of all muscle nuclei) of undifferentiated and quiescent mono-nucleated cells that reside between the basal lamina and the surface of skeletal muscle cells; a microenvironment referred to as their niche (*Cardasis and Cooper, 1975*; *Mauro, 1961*). SCs express the transcription factor Pax7 (*Seale et al., 2000*). Lineage-tracing experiments unequivocally demonstrate that Pax7-expressing (Pax7[+]) SCs are of somitic origin (*Lepper and Fan, 2010*), and they are the major contributing source of myonuclei in both postnatal development and injury-induced regeneration (*Lepper et al., 2009*). Ablation of the Pax7[+] cell population in adult mice establishes that they are essential for muscle regeneration (*Lepper et al., 2011*; *McCarthy et al., 2011*; *Murphy et al., 2011*; *Sambasivan et al., 2011*). Pax7[+] SCs first appear under the basal lamina at late stages of fetal myogenesis (day 15.5 of gestation in the mouse) (*Gros et al., 2005*; *Relaix et al., 2005*; *Schienda et al., 2006*). After birth, they remain highly active to supply myonuclei for muscle growth but gradually cease to differentiate by 3–4 weeks (*Lepper et al., 2009*; *White et al., 2010*). Therefore, SCs appear to be set-aside as a quiescent pool of stem cells during this time frame for muscle homeostasis and regeneration throughout lifetime.

Tremendous insights into the molecular regulation of muscle stem cells during adult regeneration and aging have been gained in recent years (*Brack and Munoz-Canoves, 2015*; *Dumont et al., 2015a*). The primary foci of those studies are regulators for SC proliferation, differentiation, and self-renewal during adult and aged muscle homeostasis and following injury. Information derived from these studies helps formulate strategies towards restoring a functionally competent SC pool to combat muscle aging and muscular dystrophies. By contrast, a significant gap exists in our current knowledge pertaining to the regulation of SCs shortly after birth when they provide myonuclei to drive tissue growth and transition into quiescence, i.e. their initial establishment. In particular, how the initial SC population is proportionally scaled with respect to the muscle size during the postnatal

period is entirely unknown. In adult homeostasis, SC numbers per muscle fiber in the mouse (*Collins et al., 2005*) or per defined myofiber length in human (*Boldrin and Morgan, 2011*) reveal very minimal inter-fiber variation. Teleologically, SC number to muscle size scaling would be most relevant to the replicative potential of SCs per given muscle tissue volume in regeneration. It seems reasonable to presume that the physiological SC/muscle ratio is developmentally set at a maximum in a given muscle group for effective regeneration later on in life. To what extent this ratio can be altered without changing muscle size or impacting regenerative potential remains unclear.

To date, loss of function studies have helped identify many necessary components for the initial fate specification and later maintenance of SCs. As such, changes in SC number in these animal models are not directly relevant to their initial scaling during the critical early postnatal period. By contrast, a 'gain-of-function' mouse model with super-numeral SCs is much desired, as its molecular dissection would much more likely enable the discovery of the molecular mechanism(s) sufficiently altering the initial SC pool size. Moreover, a robust SC hyperplasia mouse model could allow addressing the relevant clinical question of whether bestowing a tissue with extra stem cells would be of benefit or of detriment to the regenerative process of skeletal muscle and in other organismal systems. Yet, currently it is unknown whether the SC pool can be increased in the absence of overall muscle tissue hypertrophy.

Here, we report a murine SC hyperplasia model and investigate the effects on regenerative myogenesis. We had generated transgenic mice that express hemagglutin-tagged (HA)-TEAD1 driven by a 6.5 Kb muscle creatine kinase (MCK) control region (*Tsika et al., 2008*). TEAD1 belongs to the TEA domain transcription factor family, which serves important functional roles in multiple embryonic contexts by activating gene expression when bound to the Hippo signaling pathway effectors Yap or Taz (*Zhao et al., 2008*). Muscle-intrinsic changes effected by TEAD1 overexpression include a prominent transition toward a slow muscle contractile phenotype (*Tsika et al., 2008*). Here, we provide comprehensive analyses of the SC compartment and make the discovery of a robust, up to 6-fold increase of quiescent SCs in TEAD1-Tg mice. As a functional consequence, TEAD1-Tg muscle regenerates at a faster rate post-injury likely owing to their increased SC number as well as the increased proliferation rates observed when the myofiber-associated SCs are activated. Remarkably, in the *mdx* mouse model for Duchenne muscular dystrophy, skeletal muscle pathology is significantly ameliorated by TEAD1-overexpression, which is likely contributed in part by an increase in *utrophin* expression. Lastly, we provide evidence implicating TEAD1-Tg skeletal muscle fibers in the regulation of mouse SC number. This work implicates a role for TEAD1-induced, myofiber-derived signaling that can scale the initial SC pool size during the perinatal period. Additionally, TEAD1 overexpression appears to alter myofiber stability in the context of dystrophic disease. Further study of this mouse model will be invaluable in elucidation of physiological pathways controlling stem cell numbers, and may prove to be an entry point to modulating SC number in clinical settings.

## Results

### TEAD1-Tg mice have normal skeletal muscle size and number, but have SC hyperplasia

A remarkable feature of skeletal muscle is its ability to adapt to changing physiologic demands via reversible modulation of tissue size and of fiber-type composition, to accommodate differential force exertion or contractile usage patterns, respectively. To this end, we previously identified a regulatory role for Tead1 in the induction of slow muscle gene expression: skeletal muscle of TEAD1-Tg mice features a transition to the slow muscle contractile phenotype (*Tsika et al., 2008*; *Vyas et al., 1999*). To further confirm this transition, we employed immunofluorescence (IF) analysis to tibialis anterior (TA) muscle cross-sections from adult TEAD1-Tg and wild type (Wt) sibling mice using antibodies against fiber-type specific myosins (*Figure 1—figure supplement 1*). Fast glycolytic myofibers express IIX myosin and make up approximately half of all TA fibers of Wt mice (*Figure 1—figure supplement 1A*). By contrast, this fiber type is absent in TEAD1-Tg TA muscle (*Figure 1—figure supplement 1B*; quantification in *Figure 1I*). Reciprocal analysis using an antibody detecting any myosin but the IIX type confirmed the complete loss of this fiber type as all TA muscle fibers stained positive in TEAD1-Tg samples (*Figure 1—figure supplement 1C–D*; quantification in *Figure 1I*). While the low percentage of slow twitch (I myosin$^+$) fibers is not affected, a more than 2-fold

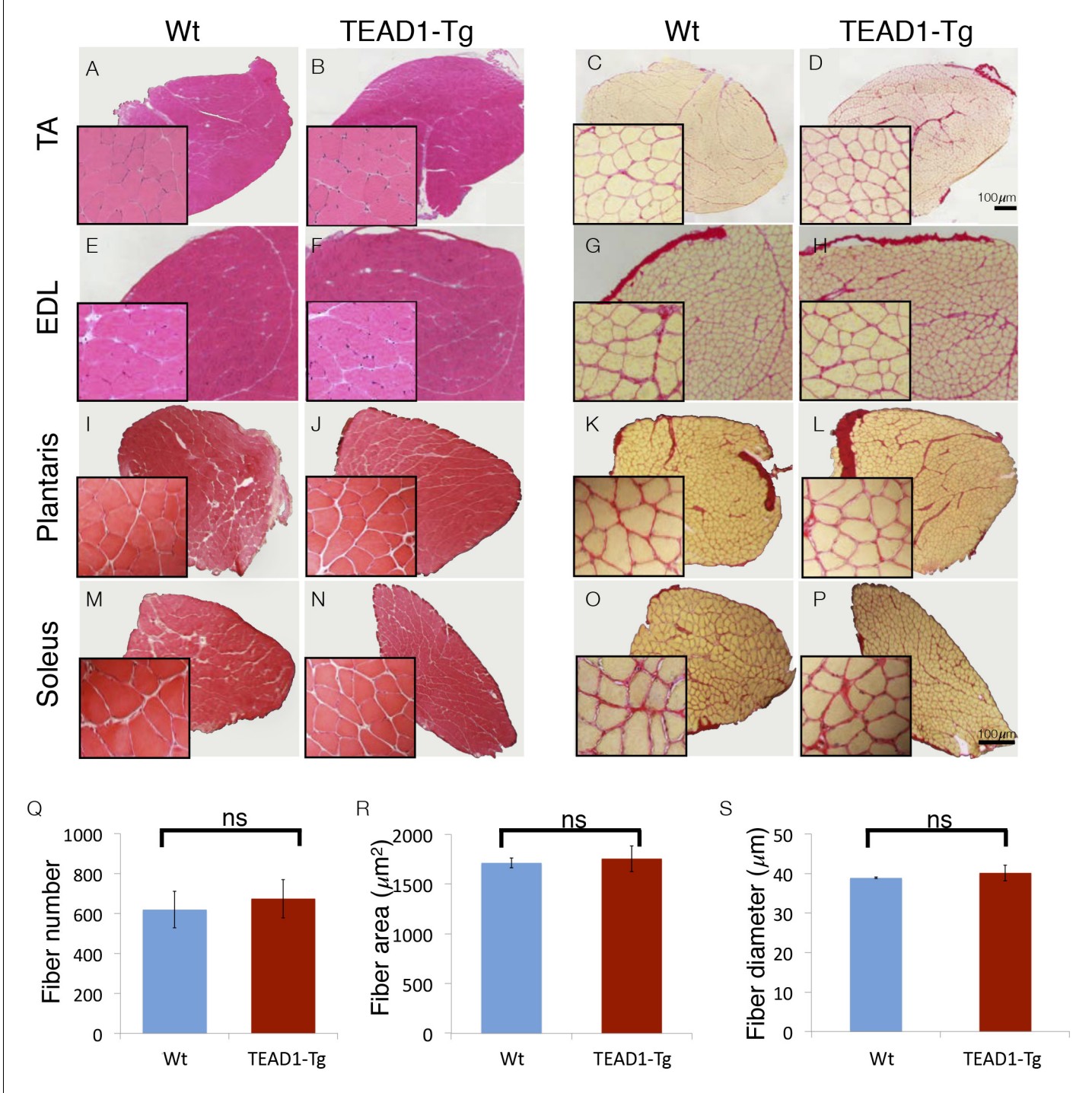

**Figure 1.** Skeletal muscle is histologically indistinguishable between Wt and TEAD1-Tg mice. (**A–P**) H and E (**A–B**, **E–F**, **I–J**, **M–N**) and Sirius Red (**C–D**, **G–H**, **K–L**, **O–P**) stains of TA (**A–D**), EDL (**E–H**), Plantaris (**I–L**), and Soleus (**M–P**) from Wt (**A, C, E, G, I, K, M, O**) and TEAD1-Tg mice (**B, D, F, H, J, L, N, P**). Insets are magnified images of the representative areas within the tissue. (**Q–S**) EDL fiber number (**Q**), fiber area (**R**), and fiber diameter (**S**) are quantified for both genotypes (n > 3 adult mice for all measurements).

The following figure supplement is available for figure 1:

**Figure supplement 1.** Ratios of type II fibers are changed in TEAD-Tg muscle.

increase in IIa myosin[+] fibers suggests that the fast glycolytic fibers are replaced in large part by fast oxidative fibers in TEAD1-Tg TA muscles (*Figure 1—figure supplement 1E–H*; quantification in *Figure 1I*). Whether additional tissue alterations accompany this TEAD1-induced change in fiber-type composition is unknown. For further characterization, we decided to evaluate muscle fiber number (hyperplasia) and size (hypertrophy) in TEAD1-Tg mice.

We investigated the size of skeletal muscle of multiple hind limb muscle groups from TEAD1-Tg mice by weight and by histological staining to determine muscle fiber size and number. Four muscle groups were chosen based on their fiber-type compositions as follows: primarily fast-twitch glycolytic fibers (EDL and plantaris), slow-twitch oxidative fibers (soleus), and mixed fast- and slow-fiber types (TA) (*Figure 1*). Histological staining by haematoxylin and eosin (H and E), as well as, by Sirius Red of the chosen muscle groups from 3-months old TEAD1-Tg and Wt littermates did not reveal histopathology as myofiber nuclei were peripherally located, and interstitial spacing between individual fibers was normal with no excess collagen or mononucleated cells (*Figure 1A–P*). TEAD1-Tg mice also did not display muscle hypertrophy, as weights of all analyzed hind limb muscles were unchanged (Table 4). Because its fibers extend the entire length of the muscle, the EDL muscle facilitates accurate determination of myofib er number and size in cross sections. No differences were detected in myofiber number (*Figure 1Q*), or in myofiber size by cross-sectional area (*Figure 1R*) and fiber diameter (*Figure 1S*) between TEAD1-Tg and Wt samples. All together, we conclude that forced expression of TEAD1 does not induce any overt pathological (i.e. fibrosis) or size (i.e. hypertrophy or hyperplasia) alterations of skeletal muscle.

Reports of a higher density of SCs in slow-twitch compared to fast-twitch muscles (*Gibson and Schultz, 1983*; *Schmalbruch and Hellhammer, 1977*) prompted us to investigate the SC compartment in TEAD1-Tg muscle, which has an increase in slow muscle gene expression (*Tsika et al., 2008*). We determined SC number by assessing Pax7[+] cells on TA muscle sections by IF staining (*Figure 2A,B[i]*; quantification in *Figure 2I*). A striking 5-fold increase in Pax7[+] cells in TEAD1-Tg was found as compared to Wt TA muscle (*Figure 2I*). This increase is of much greater magnitude than expected from the ~2-fold difference in SC number between slow- and fast-twitch muscles (*Gibson and Schultz, 1983*; *Schmalbruch and Hellhammer, 1977*). The increase in SC number was consistent between 3 independent TEAD1-Tg mouse lines (L12, L4, L14) excluding the possibility of transgene integration sites accounting for the observed increase in SCs (*Figure 2I*). For the rest of this study, we utilized line L12 exclusively for consistency.

The absence of myofiber hyperplasia and/or hypertrophy (*Figure 1*) argues for a specific SC hyperplasia in TEAD1-Tg muscle. To confirm this, we quantified myofiber nuclei in TA muscles, which revealed no differences in the myonuclei to myofiber ratio, yet the SC to myonuclei ratio is increased 5-fold when comparing TEAD1-Tg and Wt samples, thus establishing a selective effect in SC hyperplasia (*Figure 2J*). We also detected significant increases in SCs among the groups of muscle of TEAD1-Tg mice, including ~4.5-fold increase in the plantaris, ~6-fold increase in the soleus, and ~2.5-fold increase in the EDL (*Figure 2C–H[i]*; quantification in *Figure 2K*). The SC hyperplasia of the predominantly slow-twitch soleus is particularly noteworthy as it argues against the increase in SCs being a simple consequence of myofiber conversion to the slow phenotype (*Tsika et al., 2008*). Collectively, these findings rule out skeletal muscle hypertrophy and hyperplasia, and demonstrate specific SC hyperplasia in TEAD1-Tg fast- and slow-twitch skeletal muscles of the lower hind limb.

## SC hyperplasia is established during early postnatal development

During muscle development, SCs gradually become quiescent by 3–4 weeks after birth (*Lepper et al., 2009*; *White et al., 2010*). Intriguingly, peak HA-TEAD1 expression is detected prior to this time, at around postnatal day 14 (PN 14d) (*Tsika et al., 2008*). Therefore, we next examined whether TEAD1-Tg muscles show SC hyperplasia during this early postnatal time period. SCs were quantified in hind limb muscles of TEAD1-Tg and Wt sibling mice at designated postnatal time points (PN 10d, PN 12d, PN 14d and PN 20d), as was done for adult muscles at three months (*Figure 2*, Materials and methods). To monitor SC proliferation during the postnatal period, we applied the thymidine analogue EdU. At PN 10d and PN 12d, we did not detect differences in the numbers of sublaminal Pax7[+] cells or in the percentages of EdU[+]/Pax7[+] cells between TEAD1-Tg and Wt sibling TA muscles (*Figure 3A,B[iii]*; quantification in *Figure 3E,G*). At PN 14d, TEAD1-Tg TA muscles had begun accumulating an ~1.5-fold greater number of Pax7[+] cells when compared to Wt TA muscles (*Figure 3E,F*). This difference became greater at PN 20d when TA muscles boasted a

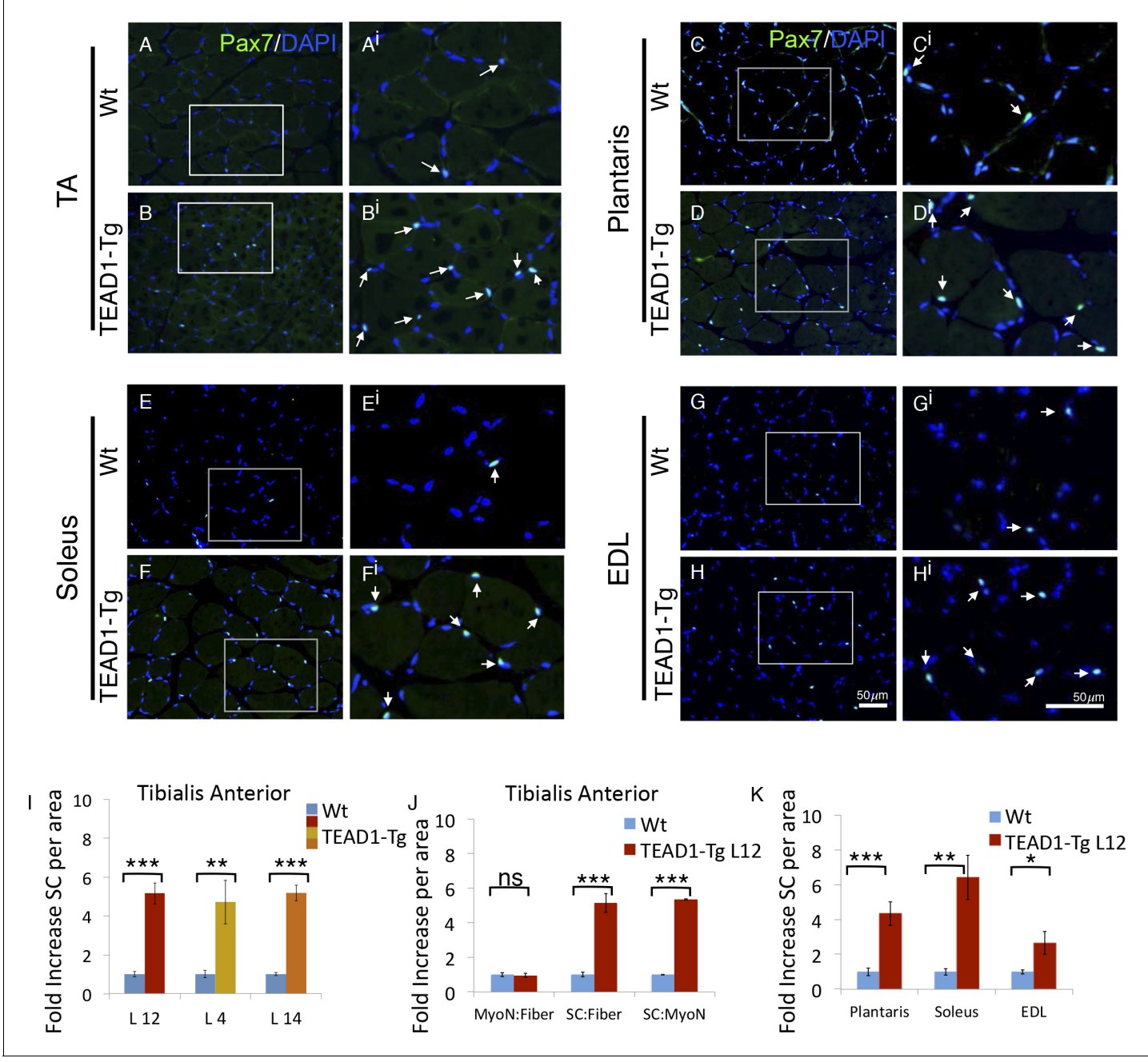

**Figure 2.** TEAD1-Tg hind limb features SC hyperplasia. (A–H$^i$) Pax7 IF in hind limb muscle groups: TA (A–A$^i$, B–B$^i$), Plantaris (C–C$^i$, D–D$^i$), Soleus (E–E$^i$, F–F$^i$), and EDL (G–G$^i$, H–H$^i$). Wt (A–A$^i$, C–C$^i$, E–E$^i$, G–G$^i$) and TEAD1-Tg muscles (B–B$^i$, D–D$^i$, F–Fi, H–H$^i$) are represented. A$^i$–H$^i$ are zoomed images of areas represented by the white boxes in A–H, arrows indicate Pax7 (green) positive nuclei (blue). (I–K) Quantification of SC hyperplasia (fold increase above Wt) in TA sections from three independent TEAD1 transgenic lines with different transgene insertion sites (I). Quantification of TA sections from Wt and TEAD1-Tg line L12 for fold increase of myonuclei per fiber, SC per fiber, and SC per myonuclei (J). Quantification of plantaris, soleus, and EDL muscle group sections for fold increase in SC number in TEAD1-Tg line L12 compared to WT (K). n > 3 adult mice for all measurements, p<0.05 represented by (*), p<0.005 represented by (**), p<0.0005 represented by (***).

>3-fold increase in SCs in TEAD1-Tg versus Wt mice (*Figure 3C,D$^{iii}$*; quantification in *Figure 3E-F*). Quantification of the percentage of sublaminal Pax7$^+$/EdU$^+$ cells at PN 20d revealed a significant 2-fold increase in the percentage of proliferative SCs in TEAD1-Tg compared to Wt TA muscles (*Figure 3G*). Conceivably, SCs could be protected from cell death in TEAD1-Tg muscle, which could

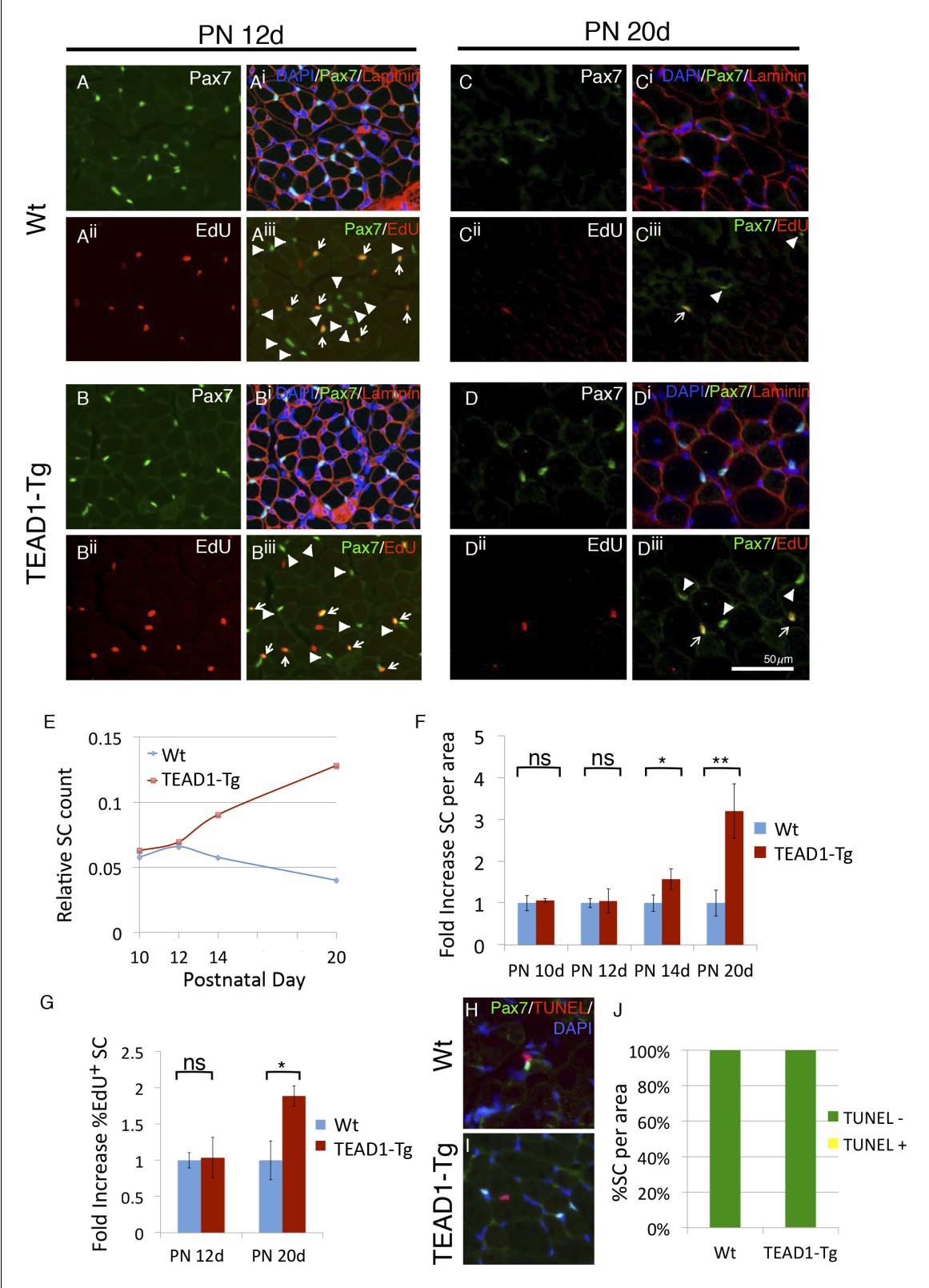

**Figure 3.** SC hyperplasia arises during perinatal stages in TEAD1-Tg mice. (A–D^iii) Wt (A–A^iii, C–C^iii) and TEAD1-Tg (B–B^iii, D–D^iii) SC numbers and proliferation were assessed. Postnatal day 12 (PN 12d, A–B^iii) and day 20 (PN 20d, C–D^iii) are represented in IF images. Pax7 (green) alone is shown in A, B, C, and D, and combined with DAPI (blue) and laminin (red) in the merged images A^i, B^i, C^i, D^i. EdU (red) alone is represented in A^ii, B^ii, C^ii, D^ii and combined with Pax7 (green) in merged images A^iii, B^iii, C^iii, D^iii, arrows indicate nuclei labeled with both EdU and Pax7 while arrowheads are Pax7

*Figure 3 continued on next page*

*Figure 3 continued*

positive nuclei that show no EdU label. (E–G) Numbers of SCs normalized to total myofibers per image is quantified for PN 10d, 12d, 14d, and 20d Wt or TEAD1-Tg TA sections (E). Fold increase in SC numbers in TEAD1-Tg TA sections compared to Wt was quantified for PN 10d, 12d, 14d, and 20d (F). Fold increase in percent EdU positive SCs in TEAD1-Tg TA sections compared to Wt was quantified for PN 12d and 20d (G). (H–J) Apoptosis rates of SCs were assessed for PN 14d. Representative images for Wt (H) and TEAD1-Tg muscle (I) show Pax7 (green) and TUNEL (red) stained nuclei (blue). Percent of SCs that were TUNEL positive or negative was quantified (J). n > 3 mice for all measurements, p<0.05 represented by (*), p<0.005 represented by (**), p<0.0005 represented by (***).

contribute to the greater number of SCs. To assay for SC apoptosis, we performed TUNEL labeling coupled with anti-Pax7 IF on TA muscle sections from TEAD1-Tg and Wt siblings at PN 14d when initial SC increases are detected (*Figure 3H,I*). A single instance of co-labeling was detected in the TEAD1-Tg sections (0.15%) while none were observed in the Wt sections (*Figure 3J*) suggesting TEAD1-Tg SC protection from apoptosis does not drive the SC hyperplasia. Together, these data indicate that TEAD1-Tg muscle accumulates super-numeral SCs beginning at ~2 weeks after birth via extending the proliferation period. Such super-numeral SCs appear to be excluded from fusion with the myofiber (no increase in the myonuclei/fiber ratio), providing the basis for the specific increase of SCs over myonuclei number.

## Super-numeral SCs have normal niche interactions

During adult homeostasis, SCs localize to the myofiber sublaminal space, are highly polarized, and maintain a quiescent state, all of which is characterized by typical molecular marker expression and a non-proliferative state. To determine if super-numeral SCs of TEAD1-Tg mice have acquired the quiescent muscle stem cell state, we performed extensive marker and proliferation analyses in adult (3-months old) mice. First, we determined proper niche localization of SCs using anti-Pax7 and anti-laminin dual-IF staining (*Figure 4A,B*). All SCs of TEAD1-Tg muscle localized to the sublaminal space (*Figure 4G*). Next, we analyzed SC polarity using β1-Integrin as a basal marker and M-Cadherin as an apical marker (*Figure 4C–F*). Similar to controls, ≥80% of SCs displayed proper polarity (*Figure 4H*). Calcitonin receptor (CTR) is specifically expressed by quiescent SCs and important for maintenance of quiescence (*Fukada et al., 2007*; *Yamaguchi et al., 2015*). Like Wt SCs, SCs from TEAD1-Tg mice express CTR (*Figure 4I–J[i]*; quantification in *Figure 4KJ*) and also the general stem cell marker, CD34 (*Figure 4K*). These data suggest that super-numeral adult SCs of TEAD1-Tg mice maintain proper cell polarity, niche interaction, as well as the quiescent state. To probe the quiescent state more rigorously, we conducted long-term proliferation assays via administering BrdU in the drinking water for a month and assaying for BrdU[+] nuclei in the sublaminal space (*Figure 4L–M*). No differences were found in proliferation between SCs of TEAD1-Tg and those of Wt muscles (*Figure 4N–O*). All together, we conclude that super-numeral SCs of TEAD1-Tg muscle acquire and maintain proper niche occupation and cell polarity, as well as, the quiescent state typical of adult muscle stem cells.

## Skeletal muscle regeneration is accelerated in TEAD1-Tg mice

Since we found SCs of TEAD1-Tg skeletal muscle to have a normal quiescent phenotype (*Figure 4*), we next asked whether they retain the functional capacity to regenerate tissue after injury. To determine this, the TA muscles of adult (2–3 months old) TEAD1-Tg and Wt littermates were injured by intra-muscular injection of cardiotoxin (CTX; see Materials and methods). Histological analyses of muscle regenerates were performed 3, 7, and 14 days post injury (dpi, *Figure 5A–F[i]*). Remarkably, while not detectable in Wt samples, very small regenerative myotubes could be detected histologically at 3 dpi in TEAD1-Tg regenerates (*Figure 5A–B[i]*), indicating that the regeneration process is accelerated in muscle with super-numeral SCs. To confirm this, we applied IF analyses using embryonic myosin heavy chain (eMyHC) as a marker for early regenerated myotubes and found widespread and robust eMyHC expression in TEAD1-Tg muscle regenerates, which is in contrast to Wt regenerates, for which eMyHC expression is more sparse and less robust (*Figure 5I–J[i]*). At 7 and 14 dpi, regenerative myofibers with centrally located myonuclei were present in both Wt and TEAD1-Tg samples, with no evidence of fibrosis or immune cell infiltration (*Figure 5C–F[i]*), demonstrating that TEAD1-Tg muscle retains full regenerative capacity. Consistent with the accelerated formation of

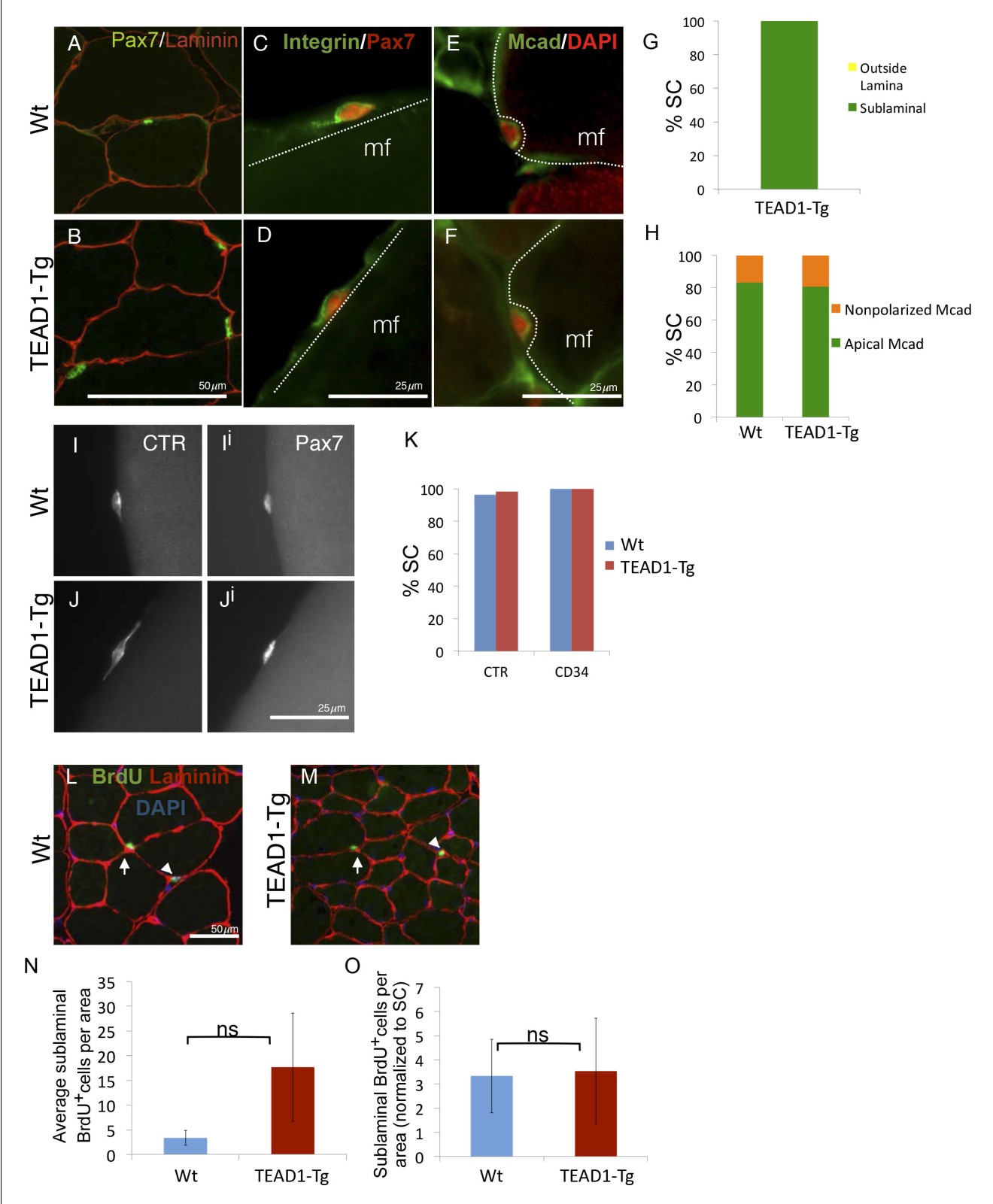

**Figure 4.** SC localization, marker expression, and long-term proliferation are no different in TEAD1-Tg compared to Wt mice. (A–H) Localization of SC (Pax7, green) within the muscle fiber basal lamina (laminin, red) shown by IF of adult Wt (A) and TEAD1-Tg (B) TA sections and quantified for TEAD1-Tg TA sections (G). Polarized Integrin-β1 expression (Integrin, green) in SCs (Pax7, red) was assessed on TEAD1-Tg and Wt isolated EDL fibers (myofiber, mf) and found to be basally localized in all instances where the position of the SC on the fiber allowed for assessment (C,D). M-Cadherin (Mcad, green)
*Figure 4 continued on next page*

*Figure 4 continued*

polarization in SCs (Pax7, nuclei shown in red) was assessed in TA sections for apical localization (**E,F**), which is quantified in **H**. Non-polarized Mcad in Wt and TEAD1-Tg samples is likely due to imperfect SC orientation within the section. **I–K**) Assessment of quiescent marker (calcitonin receptor, CTR) expression in SCs (Pax7, I[i] and J[i]) on Wt (**I**) and TEAD1-Tg (**J**) isolated EDL fibers are represented by IF and quantified in **K** along with the general stem cell marker, CD34. **L–N**) Following a month-long BrdU treatment, Wt (**L**) and TEAD1-Tg (**M**) adult TA sections were assessed for long-term proliferation. Since myonuclei are non-proliferative and SC nuclei present the only other sublaminal nuclear species, this assay cumulatively captures any SC proliferation over the one-month long period. Very low numbers of BrdU nuclei were detected in both sublaminal and interstitial compartments of TA muscles from TEAD1-Tg and Wt mice. BrdU (green) labeled nuclei (blue) within the basal lamina (red) of the myofiber were quantified (**N**) and normalized to SC number (**O**). For **G**, **H**, and **K** n > 50 cells, while for **N**, 3 mice were quantified for each genotype.

new myofibers at 3 dpi, when we quantified the sizes of regenerated myofibers at 7 dpi, we found both fiber area and diameter to be significantly increased in TEAD1-Tg muscle regenerates compared to Wt (*Figure 5G–H*). By 14 dpi, no difference in fiber size was detected. Maturation of regenerative myofibers is normal in TEAD1-Tg TA muscles as fiber sizes were no different compared to Wt at 35 dpi (*Figure 5G–H*). These data demonstrate that TEAD1-Tg muscle features accelerated kinetics of initial myotube formation upon injury. Despite this accelerated regeneration process and super-numeral SCs, we are surprised that no myofiber hypertrophy resulted from the injury-induced regenerative myogenesis. We therefore suggest that these two processes are separable.

We applied an additional injury paradigm via the myotoxin $BaCl_2$, which is thought to display less toxicity towards SCs compared to cardiotoxin (*Boldrin et al., 2012*; *Gayraud-Morel et al., 2009*), to more accurately elucidate the rapid kinetics of initial muscle repair (3, 5 and 7 dpi; *Figure 6A–H[i]*). Confirming our observations made with the cardiotoxin injury paradigm (*Figure 5*), we found significantly larger myofibers in TEAD1-Tg compared to Wt control regenerates demonstrated by increased fiber area (*Figure 6I*) and diameter (*Figure 6J*) at both 5 and 7 dpi. We conducted IF analyses of myogenic marker expression to examine the timing of myogenic differentiation in more detail (*Figure 6K–R*). Myogenin is a terminal marker of myogenic differentiation and its expression gradually increases from 3 to 5 dpi; while eMyHC expression typically initiates at 3 dpi, peaks at 4 and 5 dpi, and then diminishes by 6 to 7 dpi, at which point it becomes replaced by adult myosin heavy chain isoforms. Consistent with the earlier detection of new myotubes by 3 dpi by histology in TEAD1-Tg compared to Wt regenerates (*Figure 5A,B*), we found both Myogenin and eMyHC to be increased at this early regeneration time point in TEAD1-Tg compared to Wt samples (*Figure 6K–N*). By 5 dpi, while the number of Myogenin[+] cells is still increased in TEAD1-Tg compared to Wt regenerates (*Figure 6O–P*), eMyHC expression is already reduced in TEAD1-Tg regenerative myofibers (*Figure 6R*) compared to the 3 dpi time point (*Figure 6N*) and the peak expression observed in Wt 5 dpi skeletal muscle regenerates (*Figure 6P*), suggesting it is already being replaced by more mature myosins. Taken together, these data demonstrate acceleration of skeletal muscle regeneration by super-numeral SCs.

## Super-numeral SCs of TEAD1-Tg muscle retain full regenerative capacity over repeated injury-induced regeneration bouts

SCs represent a self-renewable reservoir of stem cells able to repeatedly provide myonuclei for muscle repair. To test if super-numeral SCs of TEAD1-Tg muscle can self-renew, we assayed for their capacity to support repeated regeneration bouts. For this, we injured the TA muscles of TEAD1-Tg and Wt littermates three times, allowing 5 weeks between injuries for complete regeneration (*Figure 7A*). Indeed, robust regeneration in both singly (*Figure 7B,D*) and triply (*Figure 7C,E*) injured TA muscles of both Wt and TEAD1-Tg mice were observed. We did not detect any excess collagen deposition in TEAD1-Tg TA muscles after one or three injury-induced regeneration bouts compared to Wt TA muscles (*Figure 7F–I*). These data demonstrate that super-numeral SCs of TEAD1-Tg skeletal muscle retain regenerative capacity after repeated injury and suggest that these SCs self-renew (to maintain their numbers as cells are recruited for fusion into myofibers). To test if these SCs participated in and contributed to the regeneration process prior to replenishing the Pax7[+] muscle reserve cell pool, we interrogated whether Pax7[+] SCs of skeletal muscle regenerates had proliferated by administering EdU during the proliferative period of the regeneration process to mark cells in S-Phase. Two weeks after injury, we could readily detect EdU[+]/Pax7[+] cells in both Wt

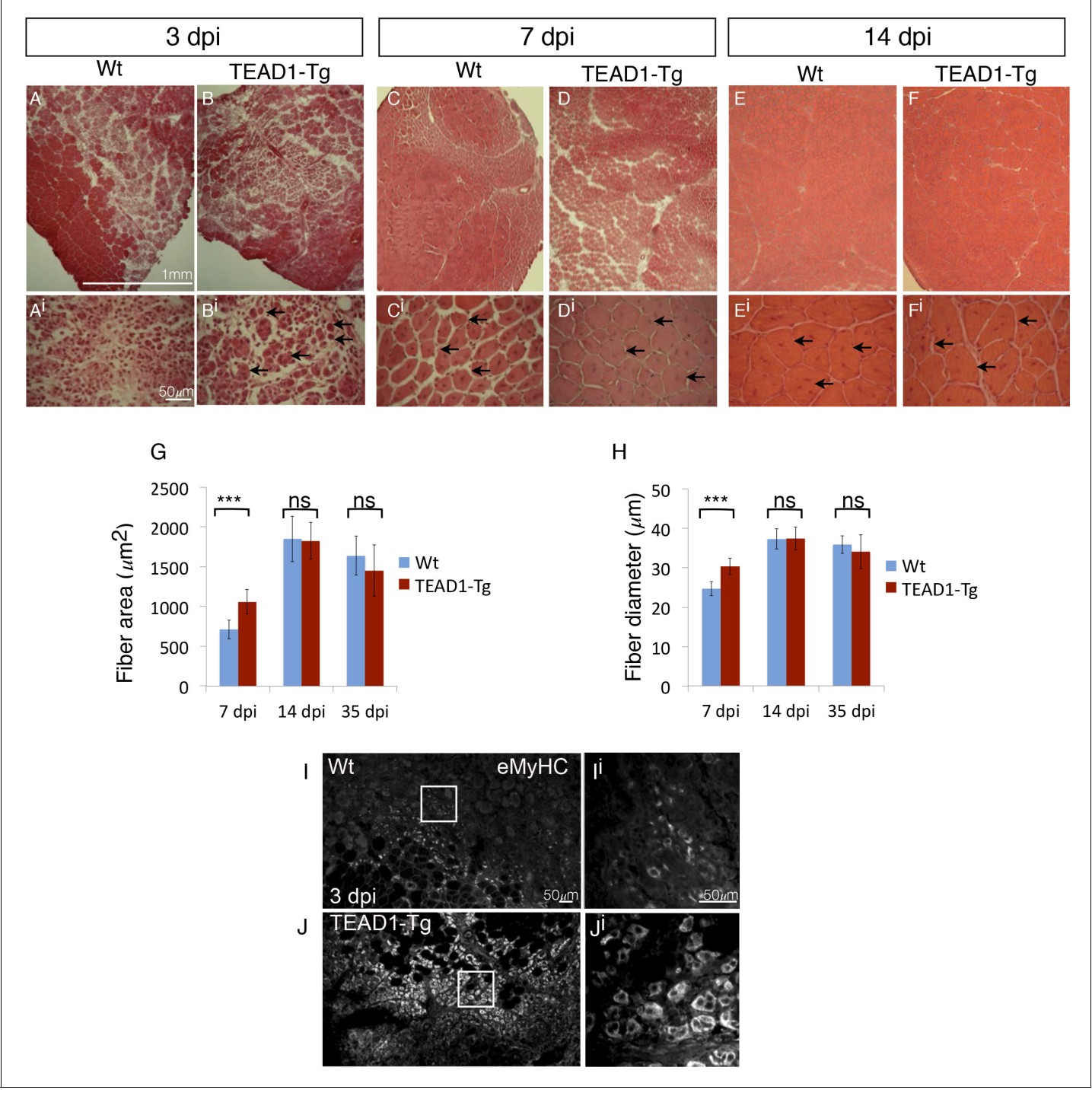

**Figure 5.** TEAD1-Tg muscle exhibits faster regeneration than Wt muscle upon injury with cardiotoxin. (A–F$^i$) H and E stains of Wt (A–A$^i$, C–C$^i$, E–E$^i$) and TEAD1-Tg (B–B$^i$, D–D$^i$, F–F$^i$) TA muscle 3 days (3 dpi, A–B$^i$), 7 days (7 dpi, C–D$^i$), or 14 days (14 dpi, E–F$^i$) after injury with cardiotoxin (CTX). Panels Ai–Fi show a higher magnification of the regenerating muscle (indicated by arrows and central nuclei). A lack of arrows in A$^i$ is due to a lack of identifiable young fibers at that stage in Wt muscle. G–H) Quantification of fiber area (G) and fiber diameter (H) for 7 dpi, 14 dpi, and 35 dpi. I–J$^i$) IF of embryonic myosin heavy chain (eMyHC, I–J$^i$) localized to regenerating fibers in Wt (I–I$^i$) and TEAD1-Tg (J–J$^i$) TA muscles 3 days after injury by CTX. Images I$^i$ and J$^i$ show a zoomed-in region indicated by the white boxes in images I and J. n > 3 mice for all samples quantified. p<0.05 represented by (*), p<0.005 represented by (**), p<0.0005 represented by (***).

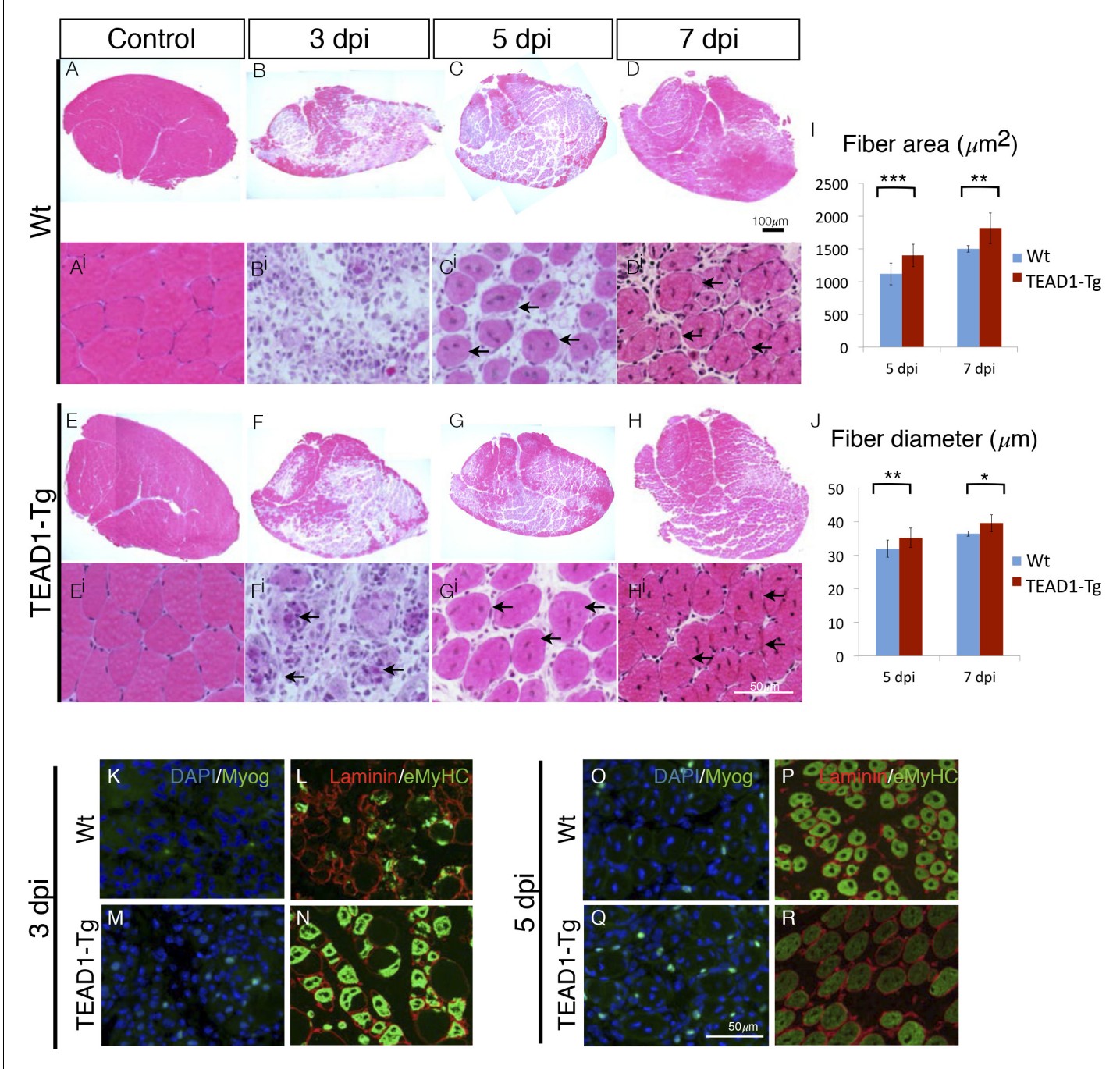

**Figure 6.** TEAD1-Tg muscle exhibits faster regeneration than Wt muscle upon injury with BaCl₂. (A–Hⁱ) H and E stains of Wt (A–Dⁱ) and TEAD1-Tg (E–Hⁱ) TA muscle 3 days (B–Bⁱ, F–Fⁱ), 5 days (C–Cⁱ, G–Gⁱ), or 7 days (D–Dⁱ, H–Hⁱ) after injury with BaCl₂ or uninjured (A–Aⁱ, E–Eⁱ). Images Aⁱ–Hⁱ are higher magnification views of regenerating areas of muscle (regenerating fibers indicated by black arrows). I–J) Quantification of fiber area (I) and fiber diameter (J) for 5 days and 7 days after BaCl₂ injury. K–R) IF images show myogenin (green) and DAPI (blue; K,M,O,Q) or embryonic myosin heavy chain (green) with laminin (red; L,N,P,R) localized to regenerating fibers in Wt (K–L, O–P) and TEAD1-Tg (M–N, Q–R) TAs 3 days and 5 days after BaCl₂ injury. For quantification of fiber number and diameter n=3 mice were used. p<0.05 represented by (*), p<0.005 represented by (**), p<0.0005 represented by (***).

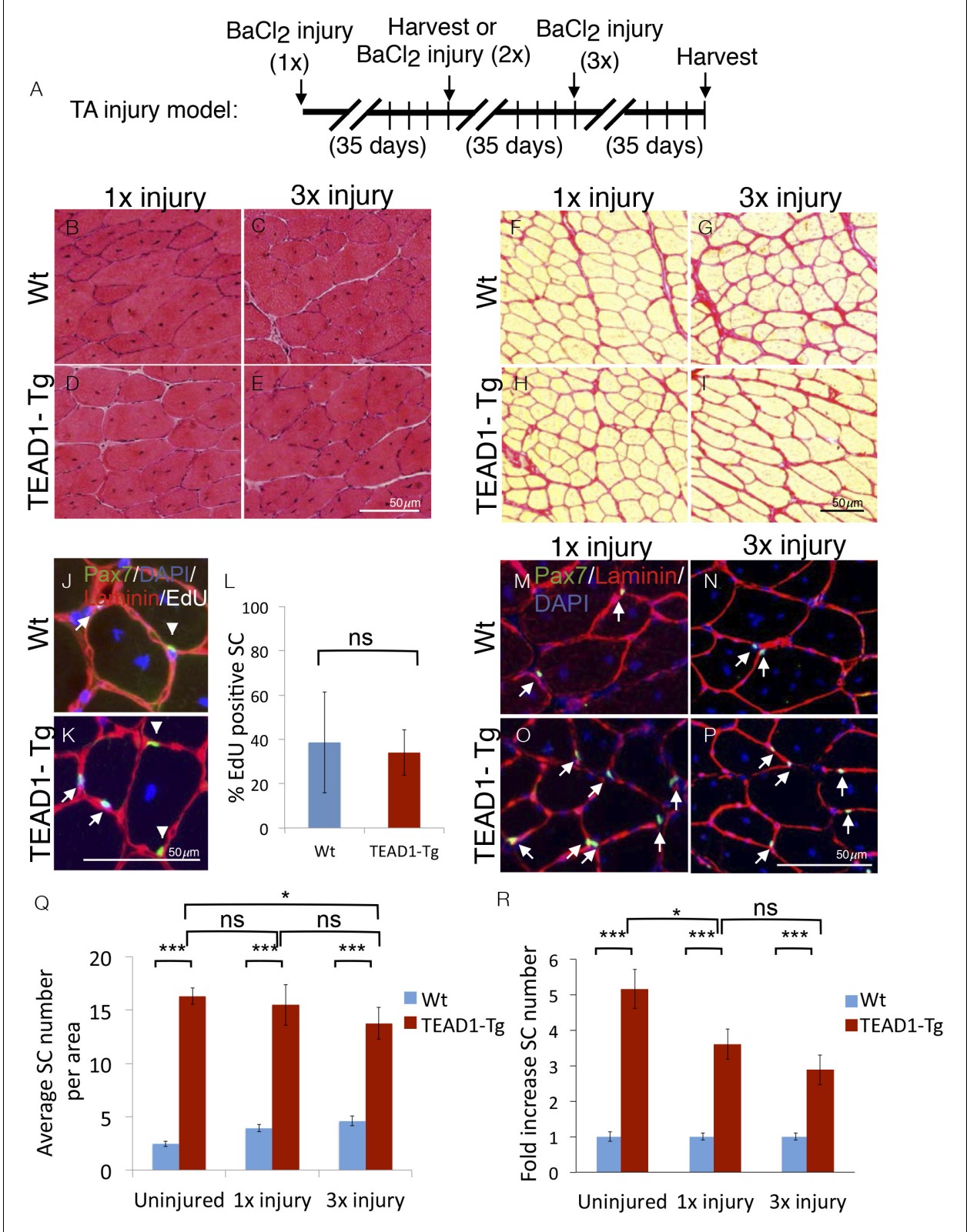

**Figure 7.** SC hyperplasia persists through multiple injuries. Adult mouse TA muscles were injured with $BaCl_2$ multiple times with 35-day regeneration periods between injuries as indicated (A). This injury paradigm applies to panels B–I, M–P) H and E stains show muscle after 35 days of regeneration following one injury (B,D) or 3 injuries (C,E) for Wt (B,C) or TEAD1-Tg muscle (D,E). (F–I) Sirius Red stains show the connective tissue after 35 days of regeneration following one injury (F,H) or 3 injuries (G,I) for Wt (F,G) or TEAD1-Tg muscle (H,I). (J–L) Wt (J) and TEAD1-Tg (K) TA muscles were injured

*Figure 7 continued on next page*

Figure 7 continued

with CTX and regenerated for 2 weeks. EdU was given on days 2–5 of regeneration. IF of Pax7 (green), laminin (red), EdU (white), and counterstained with DAPI (blue) allowed for quantification of sublaminal Pax7 and EdU positive cells (L). Myonuclei in images are EdU positive but reduced in intensity due to larger nuclear volume. M–P) Numbers of Pax7 expressing cells were assessed by IF of Pax7 (green) and laminin (red), counterstained with DAPI (blue) after 35 days of regeneration following one injury (M,O) or 3 injuries (N,P) for Wt (M,N) or TEAD1-Tg muscle (O,P) and is quantified as SC averages per area (Q) and fold increase relative to Wt (R). Uninjured data from *Figure 2* displayed for comparison. n > 3 adult mice for all measurements, p<0.05 represented by (*), p<0.005 represented by (**), p<0.0005 represented by (***).

and TEAD1-Tg samples (*Figure 7J–K*). We did not detect any statistically significant differences in the percentages of EdU$^+$/Pax7$^+$ SCs between Wt and TEAD1-Tg samples (*Figure 7L*). These data further support that SCs of TEAD1-Tg mice are self-renewing. Furthermore, we not only detected sublaminal Pax7$^+$ SCs in singly-, but also in triply-injured TA muscles of both TEAD1-Tg and Wt littermates (*Figure 7M–P*). Quantification of the number of Pax7$^+$ SCs revealed that super-numeral SCs are maintained in skeletal muscle regenerates of TEAD1-Tg mice (*Figure 7Q–R*), though the magnitude of the typical 5-fold increase in SCs in uninjured muscle is reduced to ~3.5-fold after one and ~3-fold after three injuries (*Figure 7R*). This reduction in the magnitude of the SC hyperplasia after three injuries could reflect a diminished self-renewal capacity. All together, these data demonstrate that super-numeral SCs of TEAD1-Tg muscle can self-renew and maintain the tissue's high capability for regeneration after repeated traumatic insults.

## Dystrophic pathology is ameliorated in *mdx*; TEAD1-Tg skeletal muscle

To challenge the regenerative capacity of TEAD1-Tg muscle further, we subjected the tissue to a state of chronic degeneration. To accomplish this, we bred TEAD1-Tg mice to the *mdx* mouse, a mouse model of Duchenne muscular dystrophy, which is the most severe form of the muscle wasting diseases. These mice lack the sarcolemmal protein dystrophin, which results in greatly destabilized muscle fibers and thus, chronically injured and regenerating muscle tissue, evidenced by the presence of many regenerative myofibers with centrally located nuclei, a hallmark pathological feature of this disease. Histological analyses revealed greatly reduced numbers of regenerative muscle fibers in *mdx*; TEAD1-Tg compared to *mdx* TA muscles of 2–3 months old littermates (*Figure 8A–B*). The percentage of fibers with central nuclei was reduced from ~55% in Wt to less than 20% in TEAD1-Tg muscles (*Figure 8C*), implying less degeneration in *mdx*; TEAD1-Tg muscle. We also noted a concomitant significant reduction in myofiber hypertrophy, a secondary pathological feature of the *mdx* dystrophy model (*Figure 8D–E*). To further probe the extent of improvement of the dystrophic phenotype, we assayed for fibrosis, another histological hallmark feature of skeletal muscle dystrophy. Trichrome stainings revealed a significant reduction in fibrotic area in *mdx*; TEAD1-Tg compared to *mdx* muscle (*Figure 8F–G*; quantification in *Figure 8H*). Dystrophic muscle fibers have an altered sarcolemmal permeability, which can be assessed via Evans Blue Dye (EBD) uptake. While large numbers of EBD positive fibers where detected in *mdx* muscle, EBD incorporation by fibers of *mdx*; TEAD1-Tg mice was largely stunted (*Figure 8I–J*; quantification in *Figure 8K*). The absence of evidence for worsened muscle pathology and by contrast, an amelioration of the dystrophy argues for SCs being functional in dystrophic TEAD1-Tg muscle. Lastly, we assayed for SCs in dystrophic muscle of *mdx* and *mdx*; TEAD1-Tg littermates (*Figure 8L–M*). Quantification of Pax7$^+$ cells revealed an ~2-fold increase in SCs in *mdx*; TEAD1-Tg mice (*Figure 8N*). While less dramatic compared to the fold increase observed in healthy muscle, the SC hyperplasia in dystrophic muscle is quite substantial considering the baseline SC numbers are already increased due to higher SC proliferation in the chronically de- and regenerating environment. The magnitude of the SC hyperplasia is potentially also affected by impaired dystrophin-deficient SCs (*Dumont et al., 2015b*).

Whether increased SCs, altered myofiber properties, or both contribute to the amelioration of the dystrophic pathology in *mdx*; TEAD1-Tg muscle is unclear. Besides differing in their contractile properties, slow-twitch muscle features higher utrophin levels compared to fast-twitch muscle (*Gramolini et al., 2001*). Utrophin can functionally substitute for dystrophin (*Rafael et al., 1998*). Since TEAD1-Tg muscle features a transition to the slow contractile muscle protein phenotype, we decided to investigate utrophin expression in *mdx*; TEAD1-Tg skeletal muscle. Quantitative PCR revealed an ~3-fold increase in utrophin expression in *mdx*; TEAD1-Tg compared to *mdx* muscle

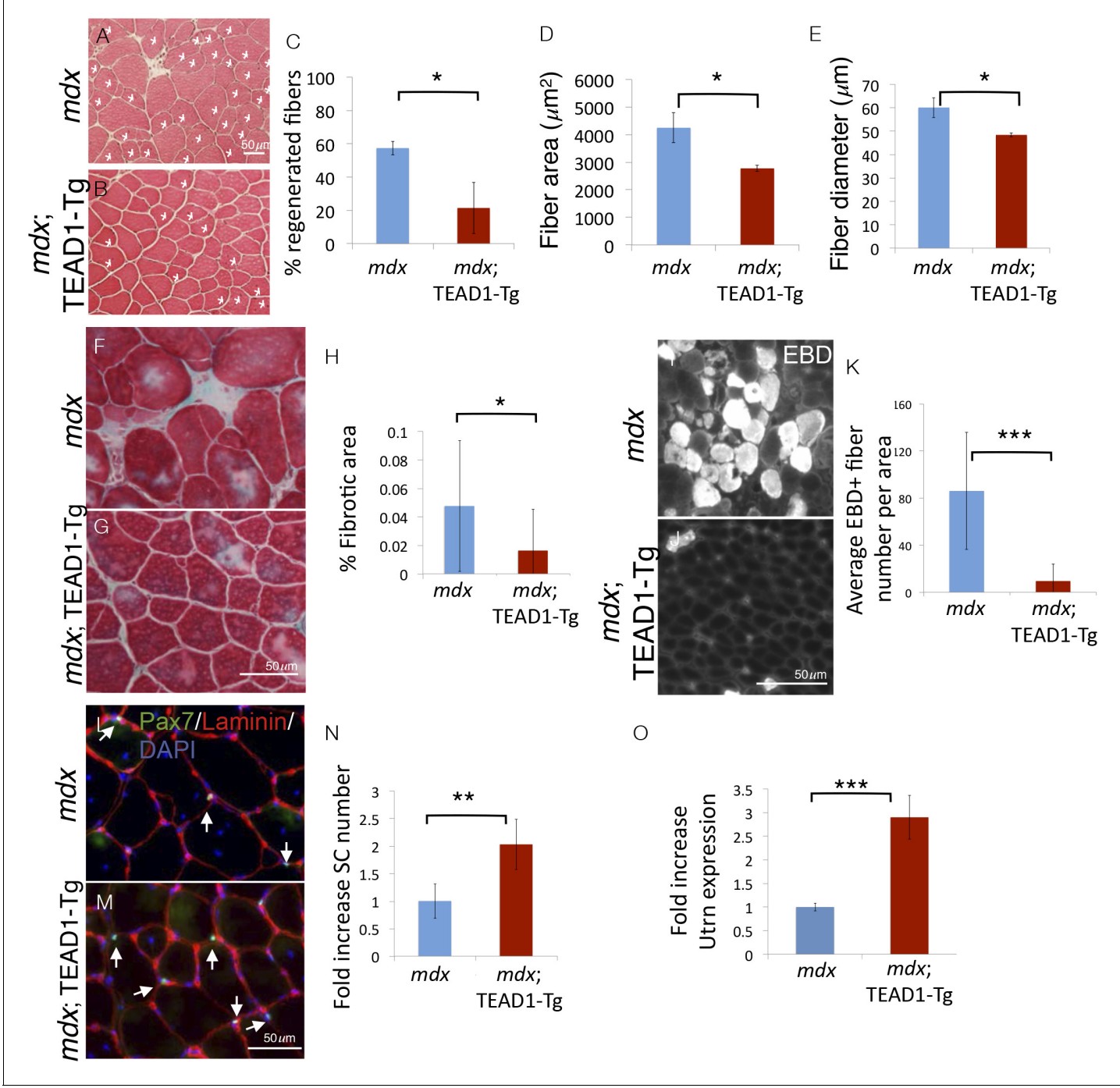

**Figure 8.** Pathologies of muscular dystrophy are ameliorated by TEAD1-Tg expression. Histology, IF analyses, and qRTPCR on dystrophic muscle from *mdx* mice modeling chronic injury. **A–C)** Asterisks in H and E stained muscle sections indicate regenerated fibers in *mdx* (**A**) and *mdx*; TEAD1-Tg mice (**B**). The percent of fibers that are regenerative is quantified for each genotype in **C**. **D–E)** The fiber area (**D**) and fiber diameter (**E**) for *mdx* or *mdx*; TEAD1-Tg TA muscle is also quantified. **F–H)** Trichrome staining was employed to label fibrotic areas in *mdx* (**F**) or *mdx*; TEAD1-Tg (**G**) TA muscle. Percent fibrotic area was quantified (**H**). **I–K)** Membrane leakage was assessed by Evan's Blue Dye incorporation into the myofibers of *mdx* (**I**) or *mdx*; TEAD1-Tg (**J**) TA muscle. Positive fibers per area were quantified (**K**). **L–N)** Numbers of Pax7 expressing cells were assessed by IF of Pax7 (green), and laminin (red), counterstained with DAPI (blue) in *mdx* (**L**) or *mdx*; TEAD1-Tg (**M**). TA muscles are quantified as fold increase relative to *mdx* in **N**. Utrophin (Utrn) expression was quantified in *mdx* or *mdx*; TEAD1-Tg TA muscle via qRTPCR (**O**). n > 3 adult mice for all measurements, p<0.05 represented by (*), p<0.005 represented by (**), p<0.0005 represented by (***).

samples (*Figure 8O*). Additionally, we assayed for revertant myofibers via anti-dystrophin IF. No increased reversion rates were found in *mdx*; TEAD1-Tg compared to *mdx* muscle samples (data not shown). These data suggest that stabilization of the sarcolemma via utrophin up-regulation contributes to amelioration of the dystrophic muscle pathology of *mdx*; TEAD1-Tg mice.

## Non-cell autonomous induction of SC hyperplasia in TEAD1-Tg muscle

The HA-tagged TEAD1 transgene is under the control of the muscle creatine kinase promoter and presumed to be exclusively expressed by the myofiber and not its associated SCs. To confirm this, we determined the expression of the TEAD1 transgene with respect to SCs or the myofiber plasmalemma by co-IF for HA and Pax7 in TEAD1-Tg skeletal muscle (*Figure 9A–B$^{iii}$*). We found no overlap of HA$^+$ and Pax7$^+$ nuclei (*Figure 9A–A$^{iii}$*). All HA expression was localized below the dystrophin$^+$ domain revealing that the only TEAD1 expressing nuclear species are those within the myofiber (*Figure 9B–B$^{iii}$*). To more rigorously probe MCK promoter-controlled TEAD1 transgene expression, we grew myoblasts in heterogeneous cultures containing both proliferative myoblasts as well as differentiated myocytes. We then assayed for transgene expression (HA) in post-mitotic differentiated myocytes (Myogenin$^+$). All HA staining was contained within the domain of Myogenin$^+$ cells (*Figure 9C*). These data demonstrate that the TEAD1 transgene is not expressed by proliferative myoblasts but only by differentiated myogenic cells. Reciprocally, we probed for TEAD1 transgene expression (via HA) in Pax7$^+$ reserve cells. As expected, transgene expression was excluded from the domain of Pax7$^+$ reserve cells (*Figure 9C*).

The above results predict that the hyper-proliferation of SCs in TEAD1-Tg skeletal muscle (*Figure 3*) depends on their associated differentiated progeny, in which the TEAD1 transgene is expressed via the MCK promoter, i.e. the myofiber. To test this prediction, we performed in vitro myoblast culture experiments either in the absence or presence of the associated TEAD1-Tg myofiber (*Figure 9D–U*). First, we cultured primary myoblasts derived from either TEAD1-Tg or Wt hind limb muscles (*Figure 9D–M*). We detected no differences in the differentiation or proliferation capacities between them. Upon serum starvation, myoblasts from either genotype readily differentiated as indicated by early (MyoD), late (Myogenin), and terminal (myosin heavy chain) differentiation marker expression (*Figure 9D–I*; quantified in *Figure 9J*). Proliferation assays using EdU incorporation, also revealed no differences between primary myoblasts from TEAD1-Tg and from Wt muscles (*Figure 9K–L*; quantified in *Figure 9M*). To determine if the TEAD1-Tg myofiber affects proliferation of its attached cohort of SCs, we cultured single myofibers from EDL muscles of TEAD1-Tg and Wt littermates and determined SC proliferation via cumulative marking of Pax7$^+$ cells in S-Phase via EdU (*Figure 9N–O*; quantification in *Figure 9P*). The majority (~92%) of SCs on Wt fibers proliferated. The proportion of proliferating SCs was significantly increased on TEAD1-Tg myofibers, as all Pax7$^+$ cells are positive for EdU (*Figure 9P*). No statistically significant differences in apoptotic rates were found via anti-Caspase-3/Pax7 co-IF staining (~0.8±1.4% versus ~3.2±1% Caspase3$^+$/Pax7$^+$ cells on Wt and TEAD1-Tg fibers, respectively; p>0.05). To determine the effect of the increased proliferation rate on SC clone formation, we quantified SCs on single myofibers at t=0 hr, and number of clones as well as clone size at t=72 hr on both Wt and TEAD1-Tg myofibers (*Figure 9Q–R*). As expected, we found significantly increased numbers of SCs associated with freshly isolated TEAD1-Tg compared to Wt myofibers (*Figure 9Q*). At t=72 hr, the number of SC clones vastly exceeded that of the starting number of SCs for both Wt and TEAD1-Tg samples suggesting that some clones split into 2 or more clones (*Figure 9Q*). Yet, significantly greater numbers of cells were found per clone on TEAD1-Tg compared to Wt fibers (*Figure 9R*). This evidence suggests that increased proliferation yields larger clones on TEAD1-Tg cultured myofibers rather than fusion of clones derived from multiple SCs, as clone fracturing occurs more frequently than clone fusion and at similar rates between Wt and TEAD1-Tg samples (*Figure 9N–R*). We next assayed cells of these clones for differentiating (MyoD$^+$), expanding (MyoD$^+$/Pax7$^+$), and self-renewing (Pax7$^+$) cell fates (*Figure 9S–T*). Quantification revealed that hyper-proliferation of SCs on TEAD1-Tg myofibers does not alter the relative distributions of these cell fates, but instead involves a proportional increase in the sizes of each fraction (*Figure 9U*). These data further confirm the SC self-renewal capacity in vivo (*Figure 7*).

A model of increased proliferation yielding a larger number of progeny, and consequently more SCs, is consistent with our in vivo postnatal proliferation rates (*Figure 3*) and explains the hyperplasia phenotype. We conclude that SC hyperplasia of TEAD1-Tg skeletal muscle is a non-cell

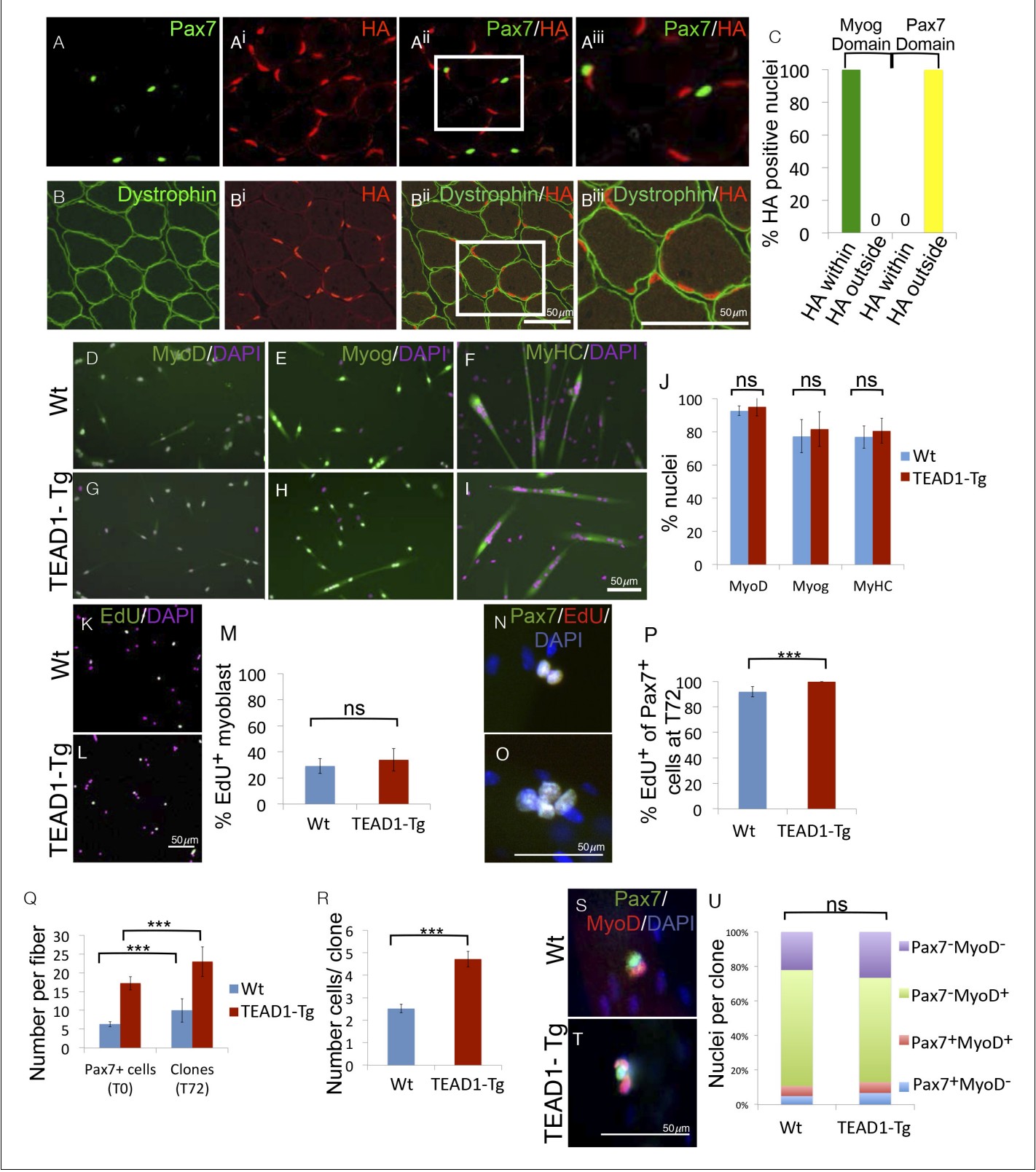

**Figure 9.** SC hyperplasia derives from cell-cell signaling between the myofiber and SCs. (**A–A**$^{iii}$) IF images of TEAD1-Tg TA adult muscle show Pax7 expressing SCs (green; **A**) and HA tagged TEAD1 (red; **A**$^i$), and are merged in image (**A**$^{ii}$) and a zoomed-in merged image (**A**$^{iii}$) to show that these genes are expressed in distinct compartments. **B–B**$^{iii}$) IF of HA-TEAD1 (red, **B**$^i$) and dystrophin (green, **B**) show that the TEAD1 transgene is expressed in nuclei of the myofiber in TA adult muscle. Transgene expression was also queried by HA colocalization with Myogenin (Myog) and Pax7-expressing

*Figure 9 continued on next page*

*Figure 9 continued*

reserve cells in differentiated cell culture experiments after 3 and 5 days of differentiation, respectively (C). (D–J) In vitro differentiation of myoblasts derived from Wt (D–F) or TEAD1-Tg (G–I) hind limb muscle shows equivalent expression of MyoD (D,G; green), Myogenin (E,H; green), and myosin heavy chain (F,I; green) and is quantified in J (n> 480 cells for all). (K–M) Cultures of Wt (K) or TEAD1-Tg (L) myoblasts show equivalent EdU label (green), which is quantified in M (n>375 cell for each). N–P) Muscle fibers and associated SC were isolated and grown in culture for 72 hr with EdU in the growth media after 24 hr in culture. A minimum number of 50 fibers were quantified. Cell clones labeled with Pax7 (green), DAPI (blue), and/or EdU (red) can be observed on isolated muscle fibers from Wt (N) or TEAD1-Tg (O) mice and are quantified in P. The average number of Pax7-positive cells (T0) and the average number of clones on muscle fibers after 72 hr (T72) in growth media are quantified in Q. Clone sizes for TEAD1-Tg and Wt fibers at 72 hr post myofiber isolation in R (n=3 mice each). After 72 hr in culture, the ratio of self-renewing SCs to differentiating cells within these clones is quantified in U (n>150 clone) for both Wt (S) and TEAD1-Tg (T) isolated fibers cultured in 10% FBS/ DMEM. p<0.05 represented by (*), p<0.005 represented by (**), p<0.0005 represented by (***).

autonomous phenomenon, which involves the coordinated scaling of the expanding, differentiating, and self-renewing myogenic cell populations.

## Discussion

Here we show a novel mouse model featuring, at its core, a robust 5- to 6-fold increase in the number of quiescent SCs, accelerated kinetics of muscle repair, amelioration of the muscular dystrophy pathology, and non-cell autonomous induction of SC hyperplasia. Because SC hyperplasia occurs without myofiber hyperplasia or hypertrophy, scaling between initial SC pool size and muscle fiber volume/length can be uncoupled, or at least altered. The teleological ramification of our findings is discussed.

### TEAD1-Tg mice: A novel mouse model for SC hyperplasia

SC hyperplasia in TEAD1-Tg mice is universal among the muscle groups analyzed, including pre-dominantly fast- or pre-dominantly slow-twitch muscle groups. Since the TEAD1-Tg features a transition to the slow-twitch muscle contractile phenotype, i.e. loss of type IIX fibers (*Figure 1—figure supplement 1*; *Tsika et al., 2008*), the pervasiveness of the SC hyperplasia among the different muscles analyzed is particularly noteworthy. Higher SC densities have been found in slow-twitch compared to fast-twitch muscles (*Gibson and Schultz, 1983*; *Schmalbruch and Hellhammer, 1977*; *Collins et al., 2005*). As such, the fast- to slow-twitch fiber transition may account for the SC hyperplasia in TEAD1-Tg mice. However, the pre-dominantly slow-twitch soleus muscle, which is devoid of type IIX fibers, features the largest increase (6-fold) in SCs in TEAD1-Tg mice. Hence, the increase in SCs cannot solely be a 'passenger effect' of the fast- to slow-twitch fiber transition. The SC hyperplasia originates at the time of peak TEAD1 transgene expression around PN 14d (*Figure 3*) and eventually features normal quiescent adult SCs indistinguishable from their Wt counterparts (*Figure 4*). In particular, we utilized this new mouse model of SC hyperplasia to investigate whether an increase in the physical number of stem cells had any impact on skeletal muscle regeneration and discovered faster regeneration kinetics in TEAD1-Tg mice (*Figures 5* and *6*). The SC hyperplasia persists through multiple and chronic injuries (*Figures 7* and *8*) and notably, derives from cell-cell signaling between the myofiber and the SCs (*Figure 9*). All together, these features uniquely prime the TEAD1-Tg mouse model for the future discovery of the elusive signaling pathway(s) regulating muscle stem cell number.

Though SC hyperplasia is not well studied, other genetic or pharmacologically induced mouse models have reported an expansion of the SC pool in adult mice. For example, skeletal muscle mutant for Collagen VI has an ~2.4-fold increase in SCs. As mutations in this gene cause myopathy or muscular dystrophy in humans (*Lampe and Bushby, 2005*), Collagen VI mutant mice also feature dystrophic muscle, which leads to SC activation as evidenced by increased proliferation and apoptosis (*Urciuolo et al., 2013*). Hence an increase in SCs is also observed in *mdx* mice (*Boldrin et al., 2009*). As such, this model features impaired regeneration and a failure to sustain the SC hyperplasia during injury-induced regenerative myogenesis (*Urciuolo et al., 2013*). Likewise, ectopic application of Wnt7a in adult regenerative muscle drives an ~2-fold expansion of proliferative SCs. No accelerated kinetics of muscle repair were noted but muscles featured both hypertrophy and hyperplasia

(*Le Grand et al., 2009*). Subsequently, Wnt7a was shown to directly act on the myofiber for induction of hypertrophy (*von Maltzahn et al., 2012a*). The expansion of SCs by Wnt7a can be further increased by the co-application with fibronectin. Whether the additional SC increase impacted myofiber hypertrophy and/or hyperplasia was not reported (*Bentzinger et al., 2013*). The source of Wnt7a in skeletal muscle is unresolved. Also, the Wnt7a-induced increase in SCs may be indirectly coupled to the myofiber hypertrophy, precluding a concrete conclusion about the signaling for the SC expansion.

While informative, the above mouse models are not useful in elucidating the molecular regulation underlying initial scaling of SC pool size during the perinatal period. They feature many distinct aspects including a less robust increase of SCs, a non-developmental origin of SC increase, overall muscle tissue hypertrophy and/or failure to generate a stably quiescent pool of adult SCs. The TEAD1 transgene induced SC hyperplasia is of the greatest known magnitude to-date, stably established during early postnatal development and features a normal adult quiescent phenotype. The timing of the SC increase in TEAD1-Tg muscle is particularly noteworthy and should incite new investigations of the aforementioned mouse models during the early postnatal period to further unravel this fundamental 'stem cell scaling' process in skeletal muscle. Furthermore, while myofiber-specific TEAD1 overexpression reveals a surprising plasticity of the muscle stem cell niche to harbor greatly increased SCs, we do not know if there are any functional requirements for the Tead transcription factor family in the regulation of SC number. Further studies of Tead loss-of-function mouse models are needed.

## Developmentally induced SC hyperplasia

The activity of early postnatal SCs is markedly different from their adult counterparts. Adult SCs are kept in a quiescent state, while early postnatal SCs are highly proliferative providing differentiated myonuclei for muscle growth as they progressively exit the cell cycle to be set aside as quiescent stem cells during the first three weeks after birth (*Lepper et al., 2009*; *White et al., 2010*). The signals that orchestrate the SC transition from the proliferative to the quiescent state are unknown but likely entail both 'proliferation' and 'quiescence' cues from the local niches leading to the formation of an adult muscle stem cell pool of defined size. Super-numeral SCs and increased SC proliferation were first detected in TEAD1-Tg muscle during this early postnatal period at 2 weeks after birth (*Figure 3*). Since TEAD1-Tg muscle is not hypertrophic and does not accumulate more nuclei than Wt (*Figures 1* and *2*), a 'Stop' signal must exist, and be still intact in TEAD1-Tg mice to prevent further fusion by excessive SCs. Thus, we propose that the extra rounds of SC proliferation are a selectively symmetric expansion of the stem cell pool. The likely mechanism driving the SC hyperplasia is over-expression or prolonged expression of physiologically normal 'proliferation' factor(s), or reduced expression of 'quiescence' factor(s) in early postnatal TEAD1-Tg muscle. Therefore, this animal model may also be useful to uncover the physiological signaling underlying the complex transition from a proliferative juvenile progenitor to a quiescent adult stem cell.

## Reduced SC hyperplasia of regenerated muscle

Applying EdU to monitor SC proliferation both in vivo (*Figure 7*) and in vitro (*Figure 9*), we found super-numeral SCs of TEAD1-Tg muscle to be self-renewing stem cells. This conclusion is further supported by the observation that the TEAD1 transgene induced SC hyperplasia is maintained even over several injury-induced regeneration cycles (*Figure 7*). Of note, the magnitude of the hyperplasia is slightly lessened with multiple bouts of regeneration. It is possible that the SC self-renewal capacity is slightly reduced in TEAD1-Tg mice. It is also possible that the regenerating tissue lacks the ability to promote SC hyperplasia to the same degree as the early postnatal myofiber. The adult niche is destroyed by injury leading to a transient loss of the TEAD1-expressing cell, i.e. the muscle fiber. This is in contrast to the 'intact' early postnatal niche, which drives the establishment of super-numeral SCs in TEAD1-Tg mice and features intimate contact between the TEAD1-expressing muscle fiber and its SCs. Moreover, there appear to be differences in the regulation of proliferating early postnatal versus quiescent adult SCs. Temporally-controlled inactivation of Pax7 in SCs shortly after birth leads to an immediate dramatic loss in regenerative capacity while inactivation in the adult leads to a slow progressive loss of SCs, which is reflected by normal regeneration over the short-term and impaired regeneration over the long-term (*Günther et al., 2013*; *Lepper et al., 2009*). It is

possible that early postnatal SCs are uniquely primed to receiving proliferation and quiescence cues from their local niches, which leads to the formation of a properly proportioned adult muscle stem cell pool. Such cues may be expressed at a higher level or for a longer period of time in the developing muscles than in adult regenerating muscles of TEAD1-Tg mice. Or, the adult SC niche may be more restrictive to proliferation, and thus limiting to the extent of re-establishment of the SC hyperplasia during regeneration. For example, the stiffness of the extracellular matrix from skeletal muscle has been demonstrated to increase with age (*Wood et al., 2014*), and substrate elasticity has been shown to affect the regenerative competence of SCs (*Gilbert et al., 2010*). Along this line of thought, it is interesting to note that neither developmental nor regenerative muscles of TEAD1-Tg mice produce hypertrophy despite the SC hyperplasia (*Figures 1*, *5* and *6*). It seems reasonable that upon acquiring the 'correct' number of myonuclei, to efficiently support the transcriptional and metabolic needs, the myofiber sends 'dominant' signal(s) to prevent additional fusion by SCs, and possibly to stop SC proliferation. A longer period of SC proliferation during early postnatal development versus adult regeneration could be the basis for the plasticity of the magnitude of the SC hyperplasia.

## Ameliorated dystrophy in *mdx*; TEAD1-Tg mice

The muscle histopathology found in the *mdx* mouse model was significantly mitigated in the TA muscle of *mdx*; TEAD1-Tg mice (*Figure 8*). This finding can likely be accounted for based on a TEAD1-induced slow-oxidative phenotype in both fast and slow-twitch muscles, which we determined previously by high-resolution electrophoretic separation and quantification of native myosin heavy chain isoforms and confirmed here by IF (*Tsika et al., 2008*). This notion is consistent with previous studies demonstrating that slow-twitch oxidative fibers in human and murine dystrophic muscles are less susceptible to the degenerative outcome of muscular dystrophies than are fast-twitch fibers (*Moens et al., 1993*; *Webster et al., 1988*; *Consolino and Brooks, 2004*). More recent therapeutic efforts shown to mitigate the dystrophic pathology in the *mdx* model have utilized transgenic (calcineurin, peroxisome proliferator-activated receptor-γ coactivator-1α, AMP-activated protein kinase, Kruppel-like factor 15) or pharmacological (Resveratol, AICAR, GW501516, Wnt7a) mediators, all previously shown to remodel dystrophic muscle to reflect a greater proportion of slow-oxidative fiber-types (*Stupka et al., 2006*; *Chakkalakal et al., 2004*; *Handschin et al., 2007*; *Hori et al., 2011*; *Tabebordbar et al., 2013*; *Ljubicic et al., 2014*; *von Maltzahn et al., 2012b*; *Morrison-Nozik et al., 2015*). It is therefore conceivable that the TEAD1-Tg-induced slow-oxidative myofiber phenotype ameliorates dystrophic pathology in *mdx* muscle in similar fashion to the aforementioned studies. In particular, utrophin, which can functionally substitute for dystrophin, is elevated in slow-twitch when compared to fast-twitch muscle (*Gramolini et al., 2001*; *Rafael et al., 1998*). Elevated *utrophin* transcript levels in *mdx*; TEAD1-Tg muscle are a highly likely contributor to the betterment of the dystrophic pathology (*Figure 8*). Additional mechanisms are likely at play. Calcineurin is a direct transcriptional activator of utrophin and skeletal-muscle specific overexpression results in increased utrophin protein levels and results in amelioration of the dystrophic pathology (*Stupka et al., 2006*). Yet, the extent of the observed amelioration is significantly smaller when compared to the *mdx*; TEAD1-Tg model. Additional properties of the myofiber affected by the TEAD1 transgene could further protect the fiber from degeneration. It is also possible that the increased number of SCs in *mdx*; TEAD1-Tg muscle contributes to a more speedy repair process, thereby stabilizing the dystrophic muscle. Or, both mechanisms may contribute. Additionally, it would be of great interest to quantify SCs in the murine models above to determine if their numbers are increased as well, which could also potentially contribute to an ameliorated disease state.

## SC hyperplasia without muscle hypertrophy

It is conceivable that the size of the resident stem cell pool could be modulated with, and thus, be directly linked to the size of tissue, for example via physical expansion or reduction of available stem cell niches. To extend this idea, it is of interest to consider the exquisite molecular mechanisms that have evolved to initially establish and then maintain proper skeletal muscle tissue size. Signaling pathways including Hippo, IGF-1, Wnt, calcium (via NFAT), and TGF-β contribute to the regulation of myofiber size and number. In particular, the TGF-β family member myostatin and its receptor activin receptor type-IIB (ActRIIB) play prominent roles, as both genetic inactivation of myostatin and

ActRIIB antagonism result in a double muscling phenotype in mice with both hyperplasia and hypertrophy (*McPherron et al., 1997*; *Zhou et al., 2010*; *Lee and McPherron, 2001*). However, it remains controversial whether a corresponding SC increase accompanies and is required for the increase in muscle mass by inhibition of myostatin signaling (*McCroskery et al., 2003*; *Wang and McPherron, 2012*; *Zhou et al., 2010*; *Lee et al., 2012*). Inactivation of ActRIIB specifically in the myofiber causes myofiber hypertrophy arguing against myofiber size regulation being dependent on the SC pool (*Lee et al., 2012*). Likewise, myostatin inhibition has been shown to result in skeletal muscle hypertrophy in the absence of satellite cell activity (*Amthor et al., 2009*). Canonical Wnt signaling has been implicated in regulating myogenic progenitor and myofiber number during fetal myogenesis and muscle repair after injury (*Hutcheson et al., 2009*; *Murphy et al., 2014*; *Rudolf et al., 2016*). Similarly, calcium signaling via NFAT proteins has been implicated in regulating skeletal muscle size (*Schulz and Yutzey, 2004*). In our previous work, we found both Wnt and NFAT signaling to be significantly reduced in TEAD1-Tg muscle (*Tsika et al., 2008*). Yet, we did not find muscle tissue size to be altered (*Figure 1*). This apparent discrepancy can likely be attributed to the late onset of MCK promoter driven TEAD1 transgene expression, which happens after muscle differentiation. Thus, neither differentiation nor fusion is affected in these mice. And the non-autonomous induction of supernumeral SCs results in the uncoupling of the molecular regulation of SC number from muscle size in TEAD1-Tg mice. Similarly, muscle groups of different ontology appear to maintain different SC to muscle nuclei ratios, i.e. branchiomeric muscles have significantly fewer SCs compared to somite-derived muscles (*Ono et al., 2010*). Still, muscle fibers and their associated cohort of SCs have co-evolved to maintain a relatively fixed ratio. While the evolutionary selection pressures for both processes are unknown they are of great biological curiosity.

# Materials and methods

## Animals

hTEAD1 transgenic mice were previously described (*Tsika et al., 2008*). Age-matched C57BL/6 siblings served as Wt controls. *mdx* mice (ID: 001801) were obtained from the Jackson Laboratory (ME). Injury with cardiotoxin (10 μM, Sigma C9759) to the TA (after anesthesia) used 50 μl for adult (>2 months) mice. For injury by $BaCl_2$ (after anesthesia), TA muscles were injected with 25 μl of 1.2% (w:v) Barium Chloride (Sigma 217565) in PBS. For multiple rounds of injury, TA muscles were injected with 25 μl of $BaCl_2$ for each injury and allowed to regenerate for 35 days between injuries. EdU (Invitrogen; CA) was administered as previously described (*Lepper et al., 2009*). BrdU (5-bromo-2'-deoxyuridine, Sigma B5002) was provided in drinking water (0.8 mg/mL) for 30 consecutive days. All procedures were approved by IACUC.

## PCR genotyping and quantitative PCR (qPCR)

For adult animals, tail DNA was used for genotyping by PCR. For perinatal mice, toe DNA was used. DNA was extracted using the ExtractN'Amp kit (Sigma XNAT2) following the manufacturer's instructions. PCR reactions were carried out using GoTaq polymerase (Promega M8291) with buffers supplied by the manufacturer with 0.1 mM dNTPs and 2.5 mM $MgCl_2$. PCR products were resolved in a 2% agarose gel, stained with 0.5 μg/mL ethidium bromide (Gibco 15585011), and digitally imaged with a Bio-Rad Gel Doc system for record keeping. Primer sequences, product sizes, and PCR conditions are in *Table 1*.

Total RNA was extracted and purified from TA muscles using the Direct-zol RNA MiniPrep Kit (ZymoResearch, R2051). Reverse transcription was performed using M-MLV Reverse Transcriptase (Thermo Scientific, 28025013). Real Time PCR was performed using the BioRad CFX96 Real-Time System and primers listed in *Table 1*. Data analysis was performed using the CFX manager. Utrophin expression was normalized to both actin and cyclophilin A expression.

## Immunofluorescence (IF)

For IF staining, muscles were isolated, partially imbedded in tragacanth (Sigma 9000-65-1) on a slice of cork, and flash frozen in isopentane (Sigma 78–78-4) cooled by liquid nitrogen. Samples were cryo-sectioned at 10 μm and mounted on Superfrost Plus slides (VWR 48311–703). Following fixation for 10 min (4% PFA/PBS) on ice and PBS wash, antigen retrieval was performed (Dako S1699). Slides

**Table 1.** PCR genotyping and RT primers, and conditions.

| Gene | Forward primer | Reverse primer | Size |
|---|---|---|---|
| TEAD-HA | 5′–ATCCATGCTTGTTACCTTCAG–3′ | 5′–ACTACAAGGACGATGACAAG–3′ | 460 bp |
| Internal Control | 5′–CAGCTCTACATCACCTGCCA–3′ | 5′–CACTGGGAAGAGACACTCAG–3′ | 520 bp |
| PCR conditions | Denature: 94°C for 30 s Anneal: 56°C for 30 s Extend: 72°C for 60 s (34 cycles) | | |
| Dystrophin (WT) | 5′–GCGCGAAACTCATCAAATATGCGTGTTAGTGT–3′ | 5′–GATACGCTGCTTTAATGCCTTTAGTCACTCAGATAGTTGAAGCCATTTTG–3′ | 134 bp |
| Dystrophin (*mdx*) | 5′–GCGCGAAACTCATCAAATATGCGTGTTAGTGT–3′ | 5′–CGGCCTGTCACTCAGATAGTTGAAGCCATTTTA–3′ | 117 bp |
| PCR conditions | Denature: 94°C for 20 s Anneal: 60°C for 20 s Extend: 72°C for 20 s (5 cycles)<br>Denature: 94°C for 20 s Anneal: 64°C for 20 s Extend: 72°C for 20 s (23 cycles) | | |
| utrophin | 5′–AGTATGGGGACCTTGAAGCC–3′ | 5′–CGAGCGTTTATCCATTTGGT–3′ | |
| cycloA | 5′–ATTTCTTTTGACTTGCGGGC–3′ | 5′–AGACTTGAAGGGGAATG–3′ | |
| actin | 5′–CCCTAAGGCCAACCGTGAA–3′ | 5′–CAGCCTGGATGGCTACGTACA–3′ | |

were placed in a solution of Sudan black dye (0.1%, Sigma 199664) in ethanol (70%) for 20 min to lessen autofluorescence. Samples were then blocked with 10% normal goat serum diluted in 0.02% triton-100/PBS (PBT). Subsequent primary and secondary antibodies were used at concentrations described (*Tables 2* and *3*) diluted in 2% FBS in 0.02% PBT for one hour to overnight in a humidified chamber. TUNEL labeling was performed immediately following the IF staining protocol using the ApopTag Red In Situ Apoptosis Detection Kit (Millipore, S7165). Coverslips were mounted with a drop of mounting media (Vector Laboratories, H1200) and sealed. All IF images were obtained using Nikon E800 and Zeiss Axioscop microscopes (described below).

## EdU and BrdU cell proliferation assays

EdU was injected at 0.1 mg/20g bodyweight. EdU was detected in muscle sections via Click-iT Kit (Invitrogen C10640). BrdU was detected following IF staining by antigen retrieval (BD biosciences 550803) and ABC amplification (Vector Laboratories PK4000). In cultured cells grown on chamber slides (Sigma C7182) coated in matrigel (Fisher CB-40234), EdU was added to a concentration of 10 µM for 45 min and detected by Click-iT Kit following IF procedures. In experiments of EdU labeled fiber-associated cell clones, EdU (10 µM) was provided throughout the culturing period beginning at t=24 hr.

## Histology

H and E staining has been previously described (*Lepper et al., 2009*). Sections were stained with Sirius Red using protocol and reagents of the Picro-Sirius Red staining kit (American MasterTech KTPSRPT) following fixation in Bouin's solution (Sigma HT01032) for 30 min at 56°C. Trichrome stains were performed according to manufacturer's instructions using the Gomori's One-Step Trichrome Kit (Polysciences Inc. 24205). All histology was imaged on the Zeiss Stemi SV11 (for low magnification) and Nikon E800 DIC microscopes (for high magnification) described below.

Evans Blue dye (EBD, Sigma E2129) was administered to mice at a 1% concentration in PBS via intra-peritoneal injection (10 µL/g bodyweight). Mice were sacrificed 20 hr later and the TA and EDL were harvested, flash frozen, then sectioned at 8 µm. Sections were fixed in cold acetone (−20°C) for 10 min, rinsed with PBS, and mounted. The dye's fluorescence was visualized via red light excitation.

## Quantifying muscle fiber size

Muscle size was determined by weight (*Table 4*) and by measurements and quantifications of muscle fibers. Cross-sections (by cryostat sections at 10 µm) of muscles were stained for dystrophin by IF or subjected to H and E stains (for *mdx* muscle samples). Using ImageJ software, fibers were manually outlined and then measured via ImageJ for area and minimum ferret diameter.

**Table 2.** Primary antibodies used for immunofluorescence. The monoclonal antibodies for Pax7, Myog, Myosin, IIx type myosin, IIa type myosin, non-IIx type myosin, I type myosin, and embryonic myosin, developed by Stefano Schiaffino, C Lucas, A Kawakami, WE Wright, DA Fischman, and HM Blau were obtained from the Developmental Studies Hybridoma Bank, created by the NICHD of the NIH and maintained at The University of Iowa, Department of Biology, Iowa City, IA 52242.

| Antibodies | Host | Dilution | Source |
|---|---|---|---|
| Anti-Pax7 | mouse (IgG1) | 1:4-5 | DSHB Pax7 (RRID:AB_528428) |
| Anti-Laminin | rabbit | 1:3000 | Sigma L9393 (RRID:AB_477163) |
| Anti-HA | rabbit | 1:1000 | Invitrogen PA1-985 (RRID:AB_559366) |
| Anti-HA | rabbit | 1:100 | Sigma H6908 (RRID:AB_260070) |
| Anti-HA | mouse (IgG1) | 1:100 | Covance mms-101P (RRID:AB_2314672) |
| Anti-B1 Integrin | rat | 1:200 | Abcam AB95623 (RRID:AB_10676803) |
| Anti-Mcadherin | mouse | 1:200 | Santa Cruz SC81471 (RRID:AB_2077111) |
| Anti-Calcitonin Receptor | rabbit | 1:250 | AbD Serotec AHP635 (RRID:AB_2068967) |
| Anti-CD34 | rat | 1:25 | BD Biosciences 553731 (RRID:AB_395015) |
| Anti-Dystrophin | mouse | 1:1000 | Genetex GTX27164 (RRID:AB_386029) |
| Anti-Dystrophin | rabbit | 1:100 | Abcam AB15277 (RRID:AB_301813) |
| Anti-MyoD | rabbit | 1:50 | Santa Cruz SC760 (RRID:AB_2148870) |
| Anti-MyoD | mouse (IgG1) | 1:200 | Santa Cruz SC32758 (RRID:AB_627978) |
| Anti-Myogenin | mouse | 1:20 | DSHB F5D (RRID:AB_528355) |
| Anti-Myogenin | mouse (IgG1) | 1:200 | Santa Cruz SC12732 (RRID:AB_627980) |
| Anti-MyHC | mouse | 1:20 | DSHB MF20 (RRID:AB_2147781) |
| Anti-eMyHC | mouse (IgG1) | 1:400 | DSHB F1.652 (RRID:AB_528358) |
| Anti-myosin (IIX) | mouse (IgM) | 1:2 | DSHB 6H1 (RRID:AB_2314830) |
| Anti-myosin (non-IIX) | mouse (IgG1) | 1:2 | DSHB BF-35 (RRID:AB_2274680) |
| Anti-myosin (IIA) | mouse (IgG1) | 1:2 | DSHB SC-71 (RRID:AB_2147165) |
| Anti-myosin (I) | mouse | 1:2 | DSHB BA-D5 (RRID:AB_2235587) |
| Anti-cleaved caspase 3 | rabbit | 1:100 | Cell Signaling 9664S (RRID:AB_2070042) |

## Myoblast isolation and IF

Genotyped animals were killed by cervical dislocation and hind limb muscles were removed and minced with scissors. Samples were digested with collagenase and dispase as described previously (*Springer et al., 2002*). Dissociated muscles were then transferred to 15 mL conical tubes with 10 mL myoblast media (20% fetal bovine serum/5% horse serum in DMEM), spun down in a clinical centrifuge, resuspended in myoblast media, and passed through a 70 µm filter. The remaining cells

**Table 3.** Secondary antibodies used for immunofluorescence.

| Host | Antigen | Fluorophore | Dilution | Source |
|------|---------|-------------|----------|--------|
| Goat | Rabbit IgG | Alexa 488 | 1:1000 | Invitrogen A11034 (RRID:AB_2576217) |
| Goat | Mouse IgG | Alexa 488 | 1:1000 | Invitrogen A11001 (RRID:AB_2534069) |
| Goat | Mouse IgG1 | Alexa 488 | 1:100 - 1:1000 | Invitrogen A21121 (RRID:AB_2535764) |
| Goat | Rabbit IgG | Alexa 568 | 1:1000 | Invitrogen A11011 (RRID:AB_2534078) |
| Goat | Rabbit IgG | Alexa 546 | 1:100 | Invitrogen A11035 (RRID:AB_2534093) |
| Goat | Mouse IgG | Alexa 568 | 1:1000 | Invitrogen A11031 (RRID:AB_2534090) |
| Goat | Rat IgG | Alexa 546 | 1:200 | Invitrogen A11081 (RRID:AB_2534125) |
| Goat | Mouse IgM | Alexa 594 | 1:1000 | Invitrogen A21044 (RRID:AB_2535713) |

were twice pre-plated onto uncoated tissue culture dishes for 30 min to remove fibroblasts. The remaining cells were quantified by hemocytometer and placed at equal numbers into wells of chamber slides (Sigma C7182) coated with matrigel (Fisher CB-40234). These cells were differentiated for 2–5 days in media containing 2% horse serum in DMEM. For IF, these cells were fixed with pre-warmed 4% PFA/PBS for 10 min, then permeabilized for 15 min in 0.5% PBT and washed with PBS. Primary and secondary antibodies were diluted in goat blocking buffer (blocking powder [Perkin Elmer P1012] in normal goat serum [Genetex GTX73245]) and applied for one hour to overnight with PBS washes in between primary and secondary antibody applications. After washing, the chamber portion was removed from the slide, and coverslips were mounted using a drop of fluoromount mounting media (Fisher OB100-01) containing DAPI.

## Isolation of myofibers

Genotyped animals were killed by cervical dislocation and EDL muscles were placed in a solution of 0.2% collagenase/DMEM for 90 min in a 37°C shaking (80 rpm) water bath. Digested muscles were placed in a pre-warmed solution of DMEM on horse serum coated tissue culture dishes for one hour. The muscles were then titurated with a large bore glass pipette to loosen outer myofibers, which were collected and moved to a new horse serum-coated dish of 10% FBS/DMEM or growth media (20% FBS/ 5% horse serum/ DMEM) via a small bore glass pipette. This continued until sufficient myofibers were obtained. Myofibers were incubated 48 or 72 hr in media (replaced daily). Myofibers were fixed in pre-warmed 4% PFA/PBS for 10 min. The myofibers were then washed three times in PBS and moved to a 1.5 mL microfuge tube. IF of isolated myofibers proceeded as follows: permeabilization (0.5% PBT) occurred for 15 min, myofibers were washed with PBS, then primary and secondary antibodies in goat blocking buffer were applied for one hour to overnight with PBS washes

**Table 4.** Weights (mgSD) of respective muscle groups in Wt vs. TEAD1-Tg mice. TEAD1-Tg hind limb muscles are equivalent in weight to Wt. Weight measurements (mg) and standard deviation for TA, EDL, Soleus, and Plantaris muscles from TEAD1-Tg or Wt mice. By t-test no significant difference was found between genotypes (n > 5 muscles for all measurements).

| | TA | EDL | Soleus | Plantaris |
|---|---|---|---|---|
| Wt | 41.0 ± 5.5 | 9.3 ± 1.9 | 8.3 ± 1.4 | 16.5 ± 2.5 |
| TEAD1-Tg | 38.6 ± 3.4 | 9.5 ± 1.5 | 9.7 ± 0.9 | 14.3 ± 2.7 |

in between primary and secondary antibody applications. Fibers were mounted on Superfrost Plus slides in a drop of mounting media containing DAPI, covered with a coverslip, and sealed with nail polish.

## Statistics

Error bars in histograms represent standard deviation (SD) over either the mean or fold change. Using an excel spreadsheet, all data was subjected to a two-tailed $t$ test or chi square test (indicated in figure legend) when appropriate to determine statistical differences. Statistical significance was defined as a p value $<0.05$. Sample size was predetermined based on published SC counts, which reveal very low variability, and on preliminary data revealing a highly robust and large change in the number of SCs in TEAD1-Tg mice.

## Microscopy

The Nikon E800 upright epifluorescence microscope uses a 100 watt mercury arc lamp fluorescent source. Images were taken with Hamamatsu Orca-Flash 4.0 LT sCMOS camera under objectives: Plan Fluor 10x (NA 0.30), 20x (NA 0.50), 40x (NA 0.75). The Zeiss Axioskop upright epifluorescence microscope uses an 89North PhotoFluor metal halide fluorescent source. Images were taken with a Zeiss AxioCam monochrome CCD camera under objectives: Plan-Neofluar 10x (NA 0.30), 20x (NA 0.50), Plan-Neofluar 40x (NA 0.75). The Zeiss Stemi SV11 dissecting microscope was used with a Canon EOS Rebel T1i DSLR camera under a 1x objective. The Nikon E800 upright microscope w/ DIC optics was used with a Canon EOS Rebel T3i DSLR camera under objectives: Plan Fluor 4x (NA 0.13), Plan Apo 10x NA 0.45 and Plan Fluor 40x (NA 1.30 Oil).

## Acknowledgement

SS and CL are supported by an NIH grant DP5OD009208 and funds from the Carnegie Institution for Science. SHL is supported by funds from the Carnegie Institution of Science. J-RK and RT are supported by AR41464. We thank Juan Ji, Christine Schramm, and Katie Capkovic-Thompson for preliminary characterization of the TEAD1-Tg mouse model. We also thank Ms. Sarina Raman for help with genotyping of mice, and Ms. Eugenia Dikovskaia and Ms. Natalia Karasseva for animal facility support. And we thank Dr. Chen-Ming Fan for critical reading of the manuscript.

## Additional information

### Funding

| Funder | Grant reference number | Author |
|---|---|---|
| NIH Office of the Director | DP5OD009208 | Sheryl Southard Christoph Lepper |
| National Institute of Arthritis and Musculoskeletal and Skin Diseases | AR41464 | Ju-Ryoung Kim Richard W Tsika |
| Carnegie Institution for Science | | Sheryl Southard SiewHui Low Christoph Lepper |

The funders had no role in study design, data collection and interpretation, or the decision to submit the work for publication.

### Author contributions

SS, Conception and design, Acquisition of data, Analysis and interpretation of data, Drafting or revising the article; J-RK, SHL, Acquisition of data, Analysis and interpretation of data; RWT, CL, Conception and design, Analysis and interpretation of data, Drafting or revising the article

### Author ORCIDs

Christoph Lepper, http://orcid.org/0000-0002-6466-0820

## Ethics

Animal experimentation: All animal experiments carried out by SS, SHL and CL were performed in accordance with protocol #138 approved by the Institutional Animal Care and Use Committee (IACUC) of the Carnegie Institution for Science (Permit number A3861-01). Y-RK and RWT performed their experiments to strict accordance of animal protocol number 07-18 approved by Missouri University Institutional Animal Care and Use Committee (MUIBC).

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
