## [Decision Letter]

Thank you for submitting your article "Muscle fiber signaling scales the myogenic stem cell pool" for consideration by *eLife*. Your article has been reviewed by two peer reviewers, and the evaluation has been overseen by Amy Wagers as the Reviewing Editor and Fiona Watt as the Senior Editor. The reviewers have opted to remain anonymous.

The reviewers have discussed the reviews with one another and the Reviewing Editor has drafted this summary with a number of requested changes and additional experiments that are deemed essential for the work to be considered for publication in *eLife*. At this point, we seek your response to these requirements to determine if it is feasible for you to comply within a reasonable length of time.

Summary:

This manuscript from the Lepper & Tsika laboratories evaluates a transgenic animal in which a muscle creatine kinase control region drives expression of TEAD1 in muscle fibers and unexpectedly results in an expansion of the muscle stem cell pool that is established towards the end of embryonic development. With exception of the conclusions regarding satellite cell self-renewal, the authors are careful to avoid overstating or over interpreting their findings, which include several novel and important observations, including: (1) that TEAD1 overexpression in muscle fibers results in a large excess of quiescent satellite cells at birth. This has really interesting implications and supports the notion that the satellite cell niche has room, under the right circumstances, and (2) that having more satellite cells can speed regeneration without causing hypertrophy or prematurely depleting the stem cell pool. Yet, there remain certain aspects of the conclusions that are less well supported by the currently available data. The reviewers engaged in a lively discussion about these issues, and while full agreement could not be achieved, consensus did emerge that certain additional studies are necessary, as detailed below.

Essential revisions:

1) Issues in interpretation of self-renewal data. All reviewers expressed concerns about the authors' interpretation of satellite cell self-renewal (subsection “Super-numeral SCs of TEAD1-Tg muscle retain full regenerative capacity over repeated injury-induced regeneration bouts”, first paragraph). To support their claim, they really need to document loss (and subsequent recovery) of satellite cell number in response to injury, together with BrdU/Edu staining. Otherwise, they cannot exclude the alternative interpretation that the reduction in fold increase (from 6-fold to 3-fold) could reflect a lack of self-renewal (due to incorporation of satellite cells into fibers and failure to regain "homeostatic" satellite cell numbers). It is also possible that there is an effect on fusion, which should be investigated through direct experimentation, such as lineage tracing. Several groups have used lineage tracing to mark SCs and demonstrate that there is ongoing SC fusion into myofibers in the adult mouse, which may be occurring in the absence of SC division. One would predict that an increase in SCs would cause an increase in SC fusion. Lineage tracing of TEAD-1 tg mice with an appropriate lineage marker is well within the expertise of the authors and could aid in further delineating the observed phenotypes. Thus, extensive additional data are needed to support the self-renewal claims, and the authors may instead wish to edit the text to remove that claim from the Results section.

2) Related to the above, in the Pax7/MyoD data in Figure 8, the authors cannot claim that Pax7^+^ only cells reflect a self-renewing fates without also showing that these the cells incorporate BrdU. They could simply be non-dividing, rather than self-renewing.

3) Issues in interpretation of *mdx* data. The amelioration of the dystrophic phenotype that the authors see is interesting and important, but it is possible that it is completely unrelated to any effect on satellite cells. TEAD overexpression could independently impact muscle fibers, e.g., by increasing their stability through upregulation of compensatory pathways (e.g. utrophin) or alteration of splicing, resulting in an increased frequency of revertant fibers. These considerations are presented in a more balanced manner in the Discussion than in the Results section, which should be rectified. It would be best if this issue were addressed experimentally, perhaps by ablating satellite cells in the model to determine whether this blunts the improvement seen with MCK-TEAD. Further data on addressing fibrosis, membrane leakage, regeneration or muscle function in *mdx*/TEAD-1 tg mice or in wild type mice would also be useful in supporting the authors' conclusions.

4) Issues related to the non-autonomy of the observed effect. Additional data in support of the cell non-autonomous mechanism is needed. It is not clear from the prior Tsika publication whether TEAD1 is expressed in satellite cells, cycling myoblasts, myocytes, or myotubes. This should be more carefully addressed experimentally to clarify expression of endogenous and exogenous TEAD at these different myogenic stages. HA staining on quiescent/homeostatic tissue, as they have done, is not sufficient. A time-course analysis of HA & TEAD1 before and after SC activation (in vivo or in vitro activation) is warranted. Presuming that HA-TEAD1 expression is lacking in satellite cells, the authors should then perform transplantation studies to confirm the non-autonomous effects of MCK-TEAD1 transgene.

5) Issues in the exclusive use of transgenic overexpression. One reviewer in particular expressed deep concerns regarding the potential for non-physiological outcomes as a result of excessive overexpression of a transcription factor. This issue should be explicitly discussed in the manuscript, and ideally, would be addressed most effectively by the inclusion of complementary loss-of-function studies for at least some of the outcomes evaluated (perhaps using AAV delivery of inhibitory constructs).

6) Further fiber type analysis, at the level of quantifying% fibers of each type, is needed in the HA-TEAD1 mice.

7) Apoptotic rates should be reported in the in vivo and in vitro systems. An alternative hypothesis is that the cells accumulate because they are protected from cell death, and this possibility is not addressed currently.

8) A deeper single fiber analysis is needed to support conclusions about SC clone size (BrdU, transparency with SC numbers, assess viability, etc.). While the authors argue that the isolated myofiber cultures (Figure 8) show increased clone sizes of SCs on TEAD-1 tg vs. wild type mice, the numbers of cells are not provided. How many SCs are on TEAD-1 myofibers at the experiment initiation and how many SCs at the conclusion? It is possible that the increase in clone size is due to segregation of TEAD-1 SCs into fewer and larger clones.

9) Finally, it is important to discuss the current data in light of the mechanisms and signaling pathways identified as perturbed in the prior work. For example, Wnt signaling is nearly eliminated as well as NFATc1 and NFATc3. Increased wnt signaling induces hypertrophy and NFAT signaling is involved in myocyte fusion, thus, TEAD-1 over expression in myofibers may inhibit cell fusion and thus, the balance of excess SCs and inhibition of hypertrophy and cell fusion might balance each other and prevent hypertrophy.

[Editors' note: further revisions were requested prior to acceptance, as described below.]

Thank you for resubmitting your work entitled "Muscle fiber signaling scales the myogenic stem cell pool" for further consideration at *eLife*. Your revised article has been favorably evaluated by Fiona Watt as the Senior Editor and two reviewers, one of whom is a member of our Board of Reviewing Editors.

The manuscript has been significantly improved but there are a few remaining issues that need to be addressed through edits to the text before acceptance. These are outlined below, accompanied by the rationale for each requested change:

1) Self-renewal data. The authors make an interesting point regarding the data of Webster et al., which indeed use intravital imaging to argue that satellite cell number is not diminished after cardiotoxin injury (although this conclusion contrasts with other reports (see Gayraud-Morel et al. 2009 and Hardy et al. 2016), and there are important limitations to their methodology, most importantly the limited muscle area that could be analyzed (only about 5-7 fiber diameters) due to limitations of microscope imaging depth (<200 μm), which could lead to sampling error if muscle damage is uneven, as is common with cardiotoxin injury). Regardless, given the authors' stance on satellite cell loss after injury, they may wish to edit the sentence in the subsection “Super-numeral SCs of TEAD1-Tg muscle retain full regenerative capacity over repeated injury-induced regeneration bouts”, which indicates that satellite cells must "replenish themselves during each regeneration bout", as this seems to suggest that satellite cell numbers drop. Perhaps they really mean "to maintain their numbers as cells are recruited for fusion into myofibers"?

Regarding the EdU incorporation data the authors have added in support of their claim of TEAD1-Tg self-renewal, these data clearly show that TEAD1-Tg cells proliferate as well as WT, but Figure 7 shows that average SC number per area increases with injury in WT and decreases in TEAD1-Tg, which suggests that TEAD1-Tg cells are maintained less well after injury, as compared to WT, although the starting number of these cells is higher. The authors consider this issue explicitly now in the Discussion, but they should also add a comment to the Results section noting that while Edu incorporation studies indicate that TEAD1-Tg satellite cells are capable of self-renewal, the loss of TEAD1-Tg cells (from uninjured to 3x injured) could reflect a diminished self-renewal capacity. The current text does not provide sufficient clarity for this important point.

2) The authors have provided an extensive discussion in their response letter delineating the reasons that they have not pursued loss-of-function studies (LOF). It is fair to say that for this family of factors, LOF studies are challenging (though maybe one should not go so far as to conclude that they are "futile"); however, the authors should still include discussion of this issue in their paper, citing their conceptual focus on "the plasticity of the muscle stem cell niche" and the need for further studies to evaluate loss of function models in order to assess the requirement for TEAD1 in normal regulation of SC number, in order to avoid misinterpretation or overinterpretation of their results by others.

3) The title should be changed; the current title is vague and furthermore implies an analysis of normal regulation of SC number (which the authors have specifically stated they do not wish to claim). A better title would be: TEAD1 overexpression in mouse muscle fibers drives satellite cell hyperplasia and counters pathological effects of dystrophin deficiency.

4) In light of the new data on utrophin expression in TEAD-expressing *mdx* muscle and since the authors do not know if the amelioration of pathology in this model is related to SC hyperplasia, the last sentence of the Abstract should be edited to include: "… targets for enhancing muscle regeneration and ameliorating muscle pathology."

5) Introduction: "Remarkably, in the *mdx* mouse model […] pathology is significantly ameliorated by TEAD1-overexpression." Because this sentence follows the discussion of supranumeral SCs in TEAD1-Tg mice, it implies that SC hyperplasia underlies the improved pathology, but the authors now provide an alternative model. The upregulation of utrophin in TEAD1-Tg *mdx* muscle should be noted here also to avoid confusion. Similarly, a following sentence: "This work implicates a role for […]" should include comment that TEAD1-induced myofiber-derived signaling can scale the SC pool AND alter myofiber stability in the context of dystrophic disease.

6) The authors have added new and rather dramatic results of fiber type analysis, which indicate a complete absence of IIx fibers in TEAD1-Tg muscle. The authors need to specify which muscle group was analyzed in Figure 1—figure supplement 1 and indicate whether these results were the same across the 4 muscle groups they chose for further analysis (Figure 1). Also, as IIx fibers account for 1/2 of all fibers in normal mice, the authors must consider the possibility (and discuss it in the manuscript) that this fiber type shift contributes to SC hyperplasia and DMD protection. Currently, this is in the Results, but not in the Discussion. Prior work (see e.g. Collins et al. 2005) clearly indicates that satellite cell content is different in muscle with different fiber type composition (e.g. Collins shows and ~4fold increase in satellite cell per myofiber in the soleus as compared to the EDL). This work should be cited and discussed.

---

## [Author Response]

Essential revisions:

*1) Issues in interpretation of self-renewal data. All reviewers expressed concerns about the authors' interpretation of satellite cell self-renewal (subsection “Super-numeral SCs of TEAD1-Tg muscle retain full regenerative capacity over repeated injury-induced regeneration bouts”, first paragraph). To support their claim, they really need to document loss (and subsequent recovery) of satellite cell number in response to injury, together with BrdU/Edu staining. Otherwise, they cannot exclude the alternative interpretation that the reduction in fold increase (from 6-fold to 3-fold) could reflect a lack of self-renewal (due to incorporation of satellite cells into fibers and failure to regain "homeostatic" satellite cell numbers). It is also possible that there is an effect on fusion, which should be investigated through direct experimentation, such as lineage tracing. Several groups have used lineage tracing to mark SCs and demonstrate that there is ongoing SC fusion into myofibers in the adult mouse, which may be occurring in the absence of SC division. One would predict that an increase in SCs would cause an increase in SC fusion. Lineage tracing of TEAD-1 tg mice with an appropriate lineage marker is well within the expertise of the authors and could aid in further delineating the observed phenotypes. Thus, extensive additional data are needed to support the self-renewal claims, and the authors may instead wish to edit the text to remove that claim from the Results section.*

We have performed additional regeneration experiments including SC proliferation assays via EdU to address the referees’ concerns about our interpretation of SC self-renewal during regeneration. We disagree with the Referee’s opinion that an initial loss of SCs in injury-induced regeneration with subsequent SC recovery needs to be documented to draw a conclusion about SC self-renewal. Indeed, intra-vital imaging of lineage-labeled SCs does not reveal any loss of SCs in response to injury (Webster et al., Cell Stem Cell, 2015). Thus, it appears such an assay would be unreliable to assessing SC self-renewal. Our new regeneration experiments including EdU show that a large percentage of SCs have proliferated prior to replenishing the SC pool in TEAD1-Tg muscle regenerates, which demonstrates their ability to self-renew (Figure 7; subsection “Super-numeral SCs of TEAD1-Tg muscle retain full regenerative capacity over repeated injury-induced regeneration bouts”. 12-20). Additional single myofiber culture experiments revealed that 100% of Pax7^+^ cells have proliferated on TEAD1-Tg myofibers (Figure 9; subsection “Non-cell autonomous induction of SC hyperplasia in TEAD1-Tg muscle”, second paragraph), further substantiating our claim of self-renewal by SCs of TEAD1-Tg muscle. We believe together, these new data provide extensive and sufficient evidence to solidify our conclusion about the self-renewal ability of SCs of TEAD1-Tg mice.

We respectfully disagree with the referees’ opinion that lineage-tracing of SCs would help to “further delineating the observed phenotypes.” Our observations center around the surprising ability of the skeletal muscle niche to support extra-numeral SCs and the effect of such super-numeral SCs on regeneration. We refrain from drawing any conclusions about the normal homeostatic maintenance of the SC hyperplasia. While we agree that future studies of the TEAD1-Tg mouse model during long-term tissue homeostasis and/or aging are much desired, they are outside the scope and focus of our present study. However, we have expanded on our discussion regarding the reduction in the fold-increase of SCs upon regeneration. Along this line, we find the Referee’s remark regarding a potential lack of self-renewal to be highly insightful. Our new data argues against a complete lack of self-renewal. However, we concede that a diminished self-renewal capacity could account for the reduction in the fold-increase of SCs after injury. The Referee’s comment forced us to think more deeply about the change in the magnitude of the SC hyperplasia. In contrast to the early post-natal SC niche that drives the establishment of super-numeral SCs in TEAD1-Tg mice, the adult niche is destroyed by injury. The transient loss of the TEAD1-expressing cell, i.e. the muscle fiber, likely diminishes the ability of the tissue to promote SC hyperplasia to the same degree as the early post-natal myofiber. We have expanded our Discussion section accordingly to account for these two possible scenarios (subsection “Reduced SC hyperplasia of regenerated muscle”).

*2) Related to the above, in the Pax7/MyoD data in Figure 8, the authors cannot claim that* Pax7^+^*only cells reflect a self-renewing fates without also showing that these the cells incorporate BrdU. They could simply be non-dividing, rather than self-renewing.*

We observe clonal expansion of SCs on TEAD1-Tg myofibers with the presence of Pax7^+^ only cells within the SC colonies. These data strongly suggest that these SCs do divide and self-renew. We made the conclusion about the SC self-renewal based on the fraction of Pax7^+^ cells per prior publications from several different laboratories. To directly demonstrate this, we have performed additional single myofiber explant experiments including SC proliferation analysis via EdU (Figure 9;subsection “Non-cell autonomous induction of SC hyperplasia in TEAD1-Tg muscle”, second paragraph). The results demonstrate that Pax7^+^ cells have indeed divided reflecting the self-renewing fate and further supporting our claim about SC self-renewal in vivo (see response to major point 1). Moreover, we found significantly increased proliferation by Pax7-expressing SCs of TEAD1-Tg compared to wild type fibers. These data corroborate the increased proliferation rates by SCs in vivo (Figure 3) and provide mechanistic insight into the acquisition of extra-numeral SCs, i.e. via hyper-proliferation.

*3) Issues in interpretation of mdx data. The amelioration of the dystrophic phenotype that the authors see is interesting and important, but it is possible that it is completely unrelated to any effect on satellite cells. TEAD overexpression could independently impact muscle fibers, e.g., by increasing their stability through upregulation of compensatory pathways (e.g. utrophin) or alteration of splicing, resulting in an increased frequency of revertant fibers. These considerations are presented in a more balanced manner in the Discussion than in the Results section, which should be rectified. It would be best if this issue were addressed experimentally, perhaps by ablating satellite cells in the model to determine whether this blunts the improvement seen with MCK-TEAD. Further data on addressing fibrosis, membrane leakage, regeneration or muscle function in mdx/TEAD-1 tg mice or in wild type mice would also be useful in supporting the authors' conclusions.*

We have performed several new experiments to strengthen our conclusions about the amelioration of the dystrophic pathology in *mdx;* TEAD1-Tg mice. We have assayed for fibrosis via Trichrome staining and membrane leakage via Evans Blue dye uptake (Figure 8;subsection “Dystrophic pathology is ameliorated in *mdx*; TEAD1-Tg skeletal muscle”, first paragraph). The results provide further evidence for a dramatically improved dystrophic pathology in these mice. To gain mechanistic insight into the amelioration of the dystrophy, we assayed for revertant fibers and changes in utrophin expression per the Referee’s insightful suggestions. While we do not detect any increased rates of myofiber reversion (subsection “Dystrophic pathology is ameliorated in *mdx*; TEAD1-Tg skeletal muscle”, last paragraph), we discovered significant up-regulation of *utrophin* transcripts in *mdx;* TEAD1-Tg muscle (Figure 8; in the aforementioned paragraph). These new data suggest that stabilization of the sarcolemma via utrophin up-regulation contributes to amelioration of the dystrophic muscle pathology of *mdx;* TEAD1-Tg mice. Please note that to accommodate extra data points our previous Figure 7 was split into two figures, i.e. Figure 7 and Figure 8.

We appreciate the referee’s opinion that our observations are “interesting and important” and agree that addressing any potential contribution from the increased SC pool to ameliorating the dystrophic phenotype experimentally is much desired. But we believe the identification of the origin of the amelioration of the dystrophy, e.g. increased SCs or increase in slow-twitch muscle program, to be deserving of an entirely separate study. To this end, prior publications reporting amelioration of dystrophy via shifting fast-twitch to slow-twitch myofibers did not investigate any potential contributions from SCs. The Referee suggested ablating SCs in our model to determine whether this blunts the improvement. We believe this to be a good experiment though it is unclear if conclusive answers can be gained from such studies. As far as we know, SC ablation in the *mdx* mouse model has not been done yet. Thus, it is not clear when the ablation can be performed and how long mice would survive. In fact, our Pax7-CreERT2 allele cannot be used for such experiments as even non-dystrophic mice die within a week after SC ablation (Development, 2011). Use of different Pax7-CreERT2 alleles could complicate the analysis, as SC ablation is incomplete. Furthermore, SC ablation in the *mdx;* TEAD1-Tg mouse would likely need to be performed at or prior to the initial onset of the SC hyperplasia 2 weeks after birth since the observed muscle improvement is a cumulative effect of myofiber-specific TEAD1 expression beginning much before the onset of the pathology in *mdx* mice. SC ablation during the early period after birth when muscle is still growing by fusion from SCs has not been attempted but may likely impact the normal growth of the muscle tissue, therefore rendering this approach impractical to obtain any conclusive data. We do agree that a separate study regarding the contribution, if any, of SCs to the improvement of the dystrophic phenotype should be done. Ideally, such a study should include all prior models based on a shift from fast- to slow-twitch muscle program and yielding amelioration of the dystrophic phenotype. We expanded our discussion about the amelioration of the dystrophic pathology in *mdx;* TEAD1-Tg mice to account for our finding of compensatory utrophin up-regulation and unknown contribution (if any) from extra-numeral SCs (subsection “Ameliorated dystrophy in *mdx*; TEAD1-Tg mice”).

*4) Issues related to the non-autonomy of the observed effect. Additional data in support of the cell non-autonomous mechanism is needed. It is not clear from the prior Tsika publication whether TEAD1 is expressed in satellite cells, cycling myoblasts, myocytes, or myotubes. This should be more carefully addressed experimentally to clarify expression of endogenous and exogenous TEAD at these different myogenic stages. HA staining on quiescent/homeostatic tissue, as they have done, is not sufficient. A time-course analysis of HA & TEAD1 before and after SC activation (in vivo or in vitro activation) is warranted. Presuming that HA-TEAD1 expression is lacking in satellite cells, the authors should then perform transplantation studies to confirm the non-autonomous effects of MCK-TEAD1 transgene.*

To address the referee’s concern regarding the non-autonomy of the SC hyperplasia, we now provide additional in vitro TEAD1 transgene expression data (Figure 9;subsection “Non-cell autonomous induction of SC hyperplasia in TEAD1-Tg muscle”, first paragraph). These new data confirm our in vivo expression analysis, which demonstrated exclusion of the HA-tagged TEAD1 from Pax7^+^ SCs. We did not detect any HA signal in Pax7-expressing reserve cells. Instead, all HA signal was exclusively contained within the domain of post-mitotic differentiated myocytes (Myogenin^+^). Together with the EdU proliferation data revealing increased proliferation of SCs from TEAD1-Tg mice only when still associated with the myofiber (Figure 9), these new data significantly substantiate our claim about the non-cell-autonomous induction of the SC increase.

Regarding the proposed transplantation studies, we believe such experiments would not yield any reliable data to demonstrate the non-cell-autonomous effects of the MCK-TEAD1 transgene. For this experiment to work, highly robust and consistently repeatable SC engraftment would need to be achieved to allow for the necessary precise quantifications of wild type SCs in TEAD1 transgenic skeletal muscle. In our hands, variability of such SC engraftment is too large to allow for this assay to work reliably. Moreover, such SC engraftments would need to be performed into early postnatal muscle at the onset of the SC hyperplasia (around 2 weeks after birth). Preliminary attempts to graft SCs into 2-week-old mice proved unsuccessful resulting in a very high rate of cannibalism. Likely for this reason, SC transplantation into early postnatal muscle has not been a standard assay in the field.

*5) Issues in the exclusive use of transgenic overexpression. One reviewer in particular expressed deep concerns regarding the potential for non-physiological outcomes as a result of excessive overexpression of a transcription factor. This issue should be explicitly discussed in the manuscript, and ideally, would be addressed most effectively by the inclusion of complementary loss-of-function studies for at least some of the outcomes evaluated (perhaps using AAV delivery of inhibitory constructs).*

We acknowledge the referee’s concerns regarding the potential for non-physiological outcomes as a result of overexpression of the TEAD1 transcription factor. While we observe a rather moderate TEAD1 overexpression levels (e.g. 3.3-fold in the gastrocnemius and 2.4-fold in the soleus), this could still result in transcriptional outcomes normally not affected by endogenous TEAD transcription factors. As such, we are careful to not draw conclusions about the role of TEAD1 in skeletal muscle fibers but instead conceptually highlight the surprising plasticity of the muscle stem cell niche to harbor a large excess of SCs concomitant with sped-up muscle regeneration. As such, we would like to argue that whether TEAD1 overexpression could affect non-physiological pathways is of lesser concern for our current study. We agree that loss-of-function studies to elucidate the role of Tead transcription factors in skeletal muscle would be highly informative. At present, such studies are hindered by the co-expression of all four functionally equivalent Tead (TEAD1,2,3,4) transcription factors by skeletal muscle tissue (Tsika et al., 2008 and references within). Furthermore, germline null mutation of TEADs 1 and 4 result in embryonic lethality, and of the small percentage of TEAD2 mice that survive, they carry defects in neural tube closure. Compound null mutations of TEAD1/TEAD2 leads to embryonic lethality. TEAD3 KO mice have not been studied as of yet. Thus, the resulting embryonic lethality of TEAD null mice preclude their use in our ongoing studies requiring the generation of conditional alleles for all four Tead genes.

The referee suggested the use of TEAD inhibitory constructs by using AAV delivery. All four TEAD-transcription factors bind to highly conserved cis-acting regulatory elements (MCAT, A/T-rich) and display overlapping functional roles (Meng et al., Genes & Devlop, 2016). To overcome issues of functional redundancy between the various TEAD family members, dominant negative TEAD constructs (i.e. dnTEAD2) that lack the TEA-DNA binding domain have been generated and tested in vitro and in vivo (e. g., Cao et al., Genes & Dev., 2008). While these constructs are successful in reducing/eliminating TEAD target gene expression, complications arise in data interpretation as the remaining C-terminal portion of TEAD is expressed and harbors the necessary interaction site for its coactivators: YAP, TAZ, and Vgll(**1-4**) ((Meng et al., Genes & Devlop, 2016). TEADs have very weak activation domains and require interactions with potent co-activators to confer robust gene expression (Meng et al., Genes & Development, 2016). The resulting data will very likely reflect altered TEAD, YAP, TAZ, and Vgll(1-4) gene programs. Ultimately, conditional KO mice for TEAD1-4 would have to be generated. We do not know if transgenic TEAD1 recruits any of its co-activators for affecting the myofiber genetic program resulting in increased SCs. Therefore, it is not clear if this approach could effectively abrogate transcriptional events stemming from TEAD1 activity. Moreover, the role of YAP in skeletal muscle fibers is controversial. While two studies suggest YAP positively regulates myofiber size (Watt et al., Nature Commun., 2015 & Goodman et al., FEBS Letter, 2015), one study implicates YAP in muscle atrophy and myopathy (Judson et al., Plos One, 2013). As such, the use of dominant negative TEAD constructs (involving blocking of YAP) appears impractical to gaining conclusive insight into the functional role of the Tead transcription factors in skeletal muscle fibers. Using gene knock-down approach via shRNA also appears impractical considering that a total of four Tead genes would have to be sufficiently knocked-down by inhibitory constructs. Moreover, virtually any cell in skeletal muscle tissue could be transduced by AAV precluding specific delivery to skeletal muscle fibers. The utility of this approach is further compromised by the rather long *lag time*after infection for maximal gene*expression*seen for AAV in virtually all tissues. This presents itself as a potentially significant problem since delivery of shRNA (or DN constructs above) would have to be performed during the early post-natal period when the quiescent SC pool is first established. With an unknown lag time of transgene expression, one cannot know when to deliver the virus. Thus, this approach is also likely futile to assessing Tead function in postnatal skeletal muscle.

*6) Further fiber type analysis, at the level of quantifying% fibers of each type, is needed in the HA-TEAD1 mice.*

We now provide fiber type analysis of TEAD1-Tg skeletal muscle via immuno-fluorescence stainings using antibodies against fiber-type specific myosins (Figure 1—figure supplement 1; subsection “TEAD1-Tg mice have normal skeletal muscle size and number, but have SC hyperplasia”, first paragraph). IIX myosin^+^ myofibers are lost and at least in part replaced by IIA myosin^+^ fibers. These results confirm the previously documented fast- to slow-twitch myofiber transition in TEAD1-Tg mice (Tsika et al., J Biol Chem, 2008).

*7) Apoptotic rates should be reported in the in vivo and in vitro systems. An alternative hypothesis is that the cells accumulate because they are protected from cell death, and this possibility is not addressed currently.*

We have performed additional experiments to assay for apoptotic rates by SCs of TEAD1-Tg muscle both in vivo and in vitro (Figure 3;subsection “SC hyperplasia is established during early postnatal development”). We did not find any differences in apoptotic rates between wild type and TEAD1-Tg samples suggesting SC protection from apoptosis is not a mechanism by which extra-numeral SCs are acquired by TEAD1-Tg muscle.

*8) A deeper single fiber analysis is needed to support conclusions about SC clone size (BrdU, transparency with SC numbers, assess viability, etc.). While the authors argue that the isolated myofiber cultures (Figure 8) show increased clone sizes of SCs on TEAD-1 tg vs. wild type mice, the numbers of cells are not provided. How many SCs are on TEAD-1 myofibers at the experiment initiation and how many SCs at the conclusion? It is possible that the increase in clone size is due to segregation of TEAD-1 SCs into fewer and larger clones.*

We have performed additional single myofiber explant experiments and quantified SCs at t=0 and SC colonies at t=72 (Figure 9;subsection “Non-cell autonomous induction of SC hyperplasia in TEAD1-Tg muscle”, second paragraph). Importantly, we did not find any evidence for fusion of SCs into fewer and larger clones to account for the increased clone sizes. Indeed, we found that clone fracturing occurs more frequently than clone fusion and at similar rates between wild type and TEAD1-Tg samples. Yet, clone size was significantly increased on TEAD1-Tg myofibers. We truly appreciate this comment by the Referee as these detailed in-depth analyses of clone size increases coupled with the EdU proliferation data (see response to major point 2) significantly strengthen our conclusions regarding TEAD1-Tg myofiber affecting SC proliferation, which provides an explanation for the hyperplasia phenotype (discussed in the last paragraph of the subsection “Non-cell autonomous induction of SC hyperplasia in TEAD1-Tg muscle”).

*9) Finally, it is important to discuss the current data in light of the mechanisms and signaling pathways identified as perturbed in the prior work. For example, Wnt signaling is nearly eliminated as well as NFATc1 and NFATc3. Increased wnt signaling induces hypertrophy and NFAT signaling is involved in myocyte fusion, thus, TEAD-1 over expression in myofibers may inhibit cell fusion and thus, the balance of excess SCs and inhibition of hypertrophy and cell fusion might balance each other and prevent hypertrophy.*

We appreciate the reference to our prior work regarding possible mechanisms and signaling pathways affecting the SC hyperplasia. We have added a discussion of our previous findings of diminished NFAT and Wnt signaling in TEAD1-Tg muscle with regard to the lack of any muscle tissue size alterations (subsection “SC hyperplasia without muscle hypertrophy”).

[Editors' note: further revisions were requested prior to acceptance, as described below.]

*1) Self-renewal data. The authors make an interesting point regarding the data of Webster et al., which indeed use intravital imaging to argue that satellite cell number is not diminished after cardiotoxin injury (although this conclusion contrasts with other reports (see Gayraud-Morel et al. 2009 and Hardy et al. 2016), and there are important limitations to their methodology, most importantly the limited muscle area that could be analyzed (only about 5-7 fiber diameters) due to limitations of microscope imaging depth (<200 μm), which could lead to sampling error if muscle damage is uneven, as is common with cardiotoxin injury). Regardless, given the authors' stance on satellite cell loss after injury, they may wish to edit the sentence in the subsection “Super-numeral SCs of TEAD1-Tg muscle retain full regenerative capacity over repeated injury-induced regeneration bouts”, which indicates that satellite cells must "replenish themselves during each regeneration bout", as this seems to suggest that satellite cell numbers drop. Perhaps they really mean "to maintain their numbers as cells are recruited for fusion into myofibers"?*

We have edited the sentence according to the referee’s suggestion (subsection “Super-numeral SCs of TEAD1-Tg muscle retain full regenerative capacity over repeated injury-induced regeneration bouts”).

*Regarding the EdU incorporation data the authors have added in support of their claim of TEAD1-Tg self-renewal, these data clearly show that TEAD1-Tg cells proliferate as well as WT, but Figure 7 shows that average SC number per area increases with injury in WT and decreases in TEAD1-Tg, which suggests that TEAD1-Tg cells are maintained less well after injury, as compared to WT, although the starting number of these cells is higher. The authors consider this issue explicitly now in the Discussion, but they should also add a comment to the Results section noting that while Edu incorporation studies indicate that TEAD1-Tg satellite cells are capable of self-renewal, the loss of TEAD1-Tg cells (from uninjured to 3x injured) could reflect a diminished self-renewal capacity. The current text does not provide sufficient clarity for this important point.*

We now include a sentence discussing the possibility of reduced self-renewal capacity by SCs of TEAD1-Tg mice in the Results section,(subsection “Super-numeral SCs of TEAD1-Tg muscle retain full regenerative capacity over repeated injury-induced regeneration bouts”).

*2) The authors have provided an extensive discussion in their response letter delineating the reasons that they have not pursued loss-of-function studies (LOF). It is fair to say that for this family of factors, LOF studies are challenging (though maybe one should not go so far as to conclude that they are "futile"); however, the authors should still include discussion of this issue in their paper, citing their conceptual focus on "the plasticity of the muscle stem cell niche" and the need for further studies to evaluate loss of function models in order to assess the requirement for TEAD1 in normal regulation of SC number, in order to avoid misinterpretation or overinterpretation of their results by others.*

We agree with the Referee that future studies are warranted to determine if the Tead family of transcription factors plays a functional role in the regulation of SC number and have included a discussion about this in the revised manuscript (subsection “TEAD1-Tg mice: A novel mouse model for SC hyperplasia”, last paragraph).

*3) The title should be changed; the current title is vague and furthermore implies an analysis of normal regulation of SC number (which the authors have specifically stated they do not wish to claim). A better title would be: TEAD1 overexpression in mouse muscle fibers drives satellite cell hyperplasia and counters pathological effects of dystrophin deficiency.*

We sincerely thank the referee or editor for suggesting a title that we feel is highly accurate for our manuscript. It is clear that a lot of thought went into crafting this title. The character count of the suggested title is 120 words without spaces, but 136 with spaces and it appears that the character limit placed by *eLife* is 120 words including spaces. We have spent a lot of efforts on our side trying to reduce the title to be within this limitation. We tried our best to not lose any information from the suggested title. We have changed our manuscript title to: “Myofiber-specific TEAD1 overexpression drives satellite cell hyperplasia and improves pathology of dystrophin deficiency”. However, we would like to note that we actually prefer the referee’s/editor’s suggested title and would be happy if it was used instead – if *eLife* allowed it.

*4) In light of the new data on utrophin expression in TEAD-expressing mdx muscle and since the authors do not know if the amelioration of pathology in this model is related to SC hyperplasia, the last sentence of the Abstract should be edited to include: "… targets for enhancing muscle regeneration and ameliorating muscle pathology."*

We thank the referee for enhancing the impact of our manuscript’s Abstract by suggesting to expand the Abstract to reflect our new insight into the amelioration of the dystrophic pathology by the TEAD1 transgene. We have edited our Abstract accordingly. Please note that to keep within less than 150 words, we have eliminated two words (‘quiescent’ and ‘adult’).

*5) Introduction: "Remarkably, in the mdx mouse model…pathology is significantly ameliorated by TEAD1-overexpression." Because this sentence follows the discussion of supranumeral SCs in TEAD1-Tg mice, it implies that SC hyperplasia underlies the improved pathology, but the authors now provide an alternative model. The upregulation of utrophin in TEAD1-Tg mdx muscle should be noted here also to avoid confusion. Similarly, a following sentence: "This work implicates a role for…" should include comment that TEAD1-induced myofiber-derived signaling can scale the SC pool AND alter myofiber stability in the context of dystrophic disease.*

We agree with the referee that the previous ordering of our summary at the end of our Introduction may have been confusing as to the origin of the amelioration of the dystrophic pathology. For enhanced clarity, have edited the text according to the referee’s suggestions (Introduction, last paragraph).

*6) The authors have added new and rather dramatic results of fiber type analysis, which indicate a complete absence of IIx fibers in TEAD1-Tg muscle. The authors need to specify which muscle group was analyzed in Figure 1—figure supplement 1 and indicate whether these results were the same across the 4 muscle groups they chose for further analysis (Figure 1). Also, as IIx fibers account for 1/2 of all fibers in normal mice, the authors must consider the possibility (and discuss it in the manuscript) that this fiber type shift contributes to SC hyperplasia and DMD protection. Currently, this is in the Results, but not in the Discussion. Prior work (see e.g. Collins et al. 2005) clearly indicates that satellite cell content is different in muscle with different fiber type composition (e.g. Collins shows and ~4fold increase in satellite cell per myofiber in the soleus as compared to the EDL). This work should be cited and discussed.*

We thank the referee for bringing to our attention that we had not sufficiently discussed potential contributions from the fast- to slow-twitch myofiber transition to the SC hyperplasia in our Discussion section. While the results of an absence of IIX fibers by immuno-fluorescence staining on sections is dramatic, they did not come as a surprise to us, as our previous work had already demonstrated this loss by high resolution glycerol gel electrophoresis (Tsika, 2008). If the SC hyperplasia was a ‘passive bystander effect’ of the fast- to slow-twitch myofiber transition, then little to no change in SC numbers should be observed in a pre-dominantly slow-twitch muscle devoid of type IIX fibers such as the soleus. The opposite, however, is the case. In TEAD1-Tg mice, the soleus features the largest increase in SCs of all muscles analyzed. These data strongly argue against the SC hyperplasia to exclusively originate from the fast- to slow-twitch myofiber transition, i.e. loss of type IIX fibers. We welcome the opportunity to make this point clearer and thank the referee for bringing work from Collins et al. to our attention, which we now also cite (subsection “TEAD1-Tg mice: A novel mouse model for SC hyperplasia”, first paragraph). We have applied the fiber type analysis to the TA muscle, which normally contains both fast- and slow-twitch myofibers. This is now indicated in the figure legend of Figure 1—figure supplement 1.